# VideoMAR: Autoregressive Video Generation with Continuous Tokens

**Hu Yu**[1]    **Biao Gong**[2*]    **Hangjie Yuan**[2]    **DanDan Zheng**[2]    **Weilong Chai**[2]

**Jingdong Chen**[2]    **Kecheng Zheng**[2]    **Feng Zhao**[1†]

[1] MoE Key Lab of BIPC, University of Science and Technology of China [2] Independent researcher

`https://github.com/inclusionAI/Ming-VideoMAR`

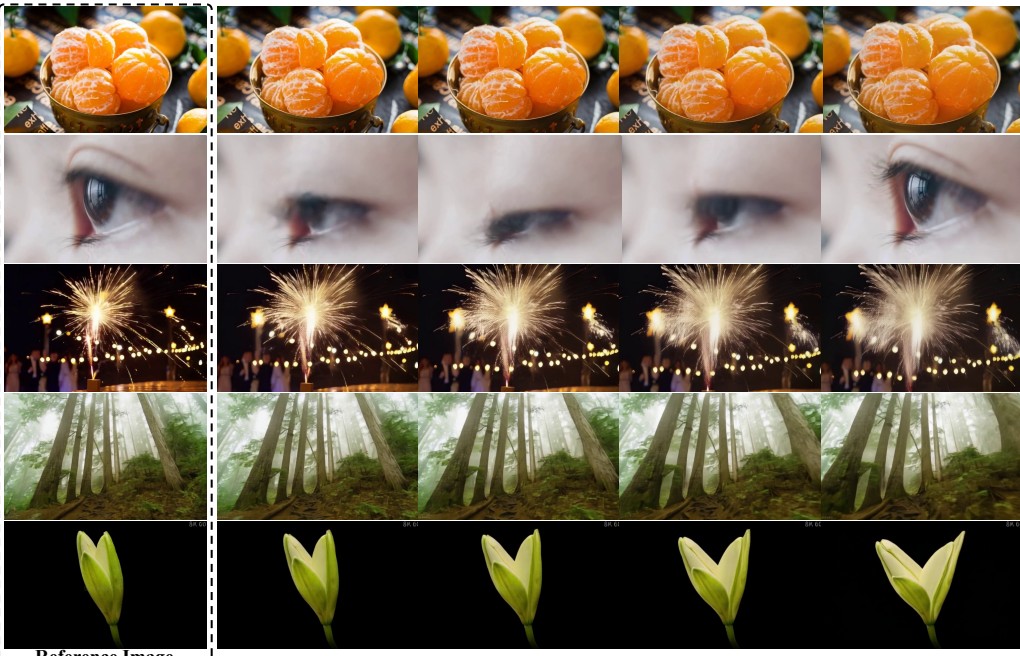

Figure 1: Autoregressive image-to-video samples of VideoMAR with continuous tokens.

## Abstract

Mask-based autoregressive models have demonstrated promising image generation capability in continuous space. However, their potential for video generation remains under-explored. In this paper, we propose **VideoMAR**, a concise and efficient decoder-only autoregressive image-to-video model with continuous tokens, composing temporal frame-by-frame and spatial masked generation. We first identify temporal causality and spatial bi-directionality as the first principle of video AR models, and propose the next-frame diffusion loss for the integration of mask and video generation. Besides, the huge cost and difficulty of long sequence autoregressive modeling is a basic but crucial issue. To this end, we propose the temporal short-to-long curriculum learning and spatial progressive resolution training, and employ progressive temperature strategy at inference time to mitigate the accumulation error. Furthermore, VideoMAR replicates several unique capacities of language models to video generation. It inherently bears high efficiency due to simultaneous temporal-wise KV cache and spatial-wise parallel generation, and presents the capacity of spatial and temporal extrapolation via

---

[*]`Project lead`
[†]`Corresponding author`

39th Conference on Neural Information Processing Systems (NeurIPS 2025).

3D rotary embeddings. On the VBench-I2V benchmark, VideoMAR surpasses the previous state-of-the-art (Cosmos I2V) while requiring significantly fewer parameters (9.3%), training data (0.5%), and GPU resources (0.2%).

# 1 Introduction

Building upon autoregressive (AR) models, large language models (LLMs) [23, 22] have unified and dominated language tasks with promising intelligence in generality and versatility, demonstrating a promising path toward artificial general intelligence (AGI). Recently, MAR series methods [19, 10, 35] have demonstrated great success of conducting autoregressive image generation in continuous space. However, its potential for autoregressive video generation remains under-explored. Compared to image, video data is temporally sequential, making it more suitable for autoregressive modeling.

A naive way of video autoregressive modeling directly adapts the paradigm of language models [22], which factorizes frames into discrete tokens and applies next-token prediction (denoted as NTP) in raster-scan order [17, 29, 1]. However, this paradigm for video generation suffers from several limitations: **1)** Discrete tokens deviate from the inherent continuous distribution of video data and irreparably induce significant information loss. **2)** The unidirectional modeling of visual tokens deviate from the inter-dependency nature of tokens within identical frame, and may be suboptimal in performance [19, 10]. **3)** NTP demands substantial inference steps for video generation.

Compared to NTP, mask-based autoregressive generation is a more promising direction [19]. However, it is nontrivial to incorporate mask mechanism into autoregressive video generation. A desired way is to sequentially generate each frame depending on all the previous context frames. While, this poses challenges for the introduction of mask. Prevailing methods [3, 36] apply mask to each frame, but introduces training-inference gap. NOVA [8] proposes to decompose the temporal and spatial generation via generating the coarse features frame-by-frame and refines each frame with a spatial layer, but complicates the framework and weakens the temporal smoothness. MAGI [39] mitigates this issue by appending a complete copy of video sequence during training, but doubles the sequence length and training cost. Therefore, mask-based video autoregressive generation, lying as a promising but challenging paradigm, still requires further exploration.

In this paper, we propose VideoMAR, a decoder-only autoregressive video generation model with continuous tokens, integrating **temporal frame-by-frame** and **spatial masked generation**. To meet the requirement of sequentially generating each frame depending on all the previous context frames, VideoMAR preserves the complete context and introduces a next-frame diffusion loss during training. Besides, the extremely long token sequences of video data poses significant challenges in both efficiency and difficulty. To this end, we propose tailored strategies for training and inference. *During training*, we propose the short-to-long curriculum learning to reduce the training difficulty and cost, and establish the two-stage progressive-resolution training to support higher resolution video generation. *During inference*, long token sequence generation is prone to suffer from severe accumulation error in late frames, due to exposure bias issue [38]. We identify that temperature plays a crucial role to eliminate this error and propose the progressive temperature strategy.

Furthermore, VideoMAR **replicates several unique capacities** of language models to video generation, *e.g.,* key-value cache and extrapolation, demonstrating the potential for multi-modal unification. For example, thanks to our design, VideoMAR inherently bears high efficiency due to simultaneous temporal-wise KV cache and spatial-wise parallel generation. VideoMAR, for the first time, also unlocks the capacity of simultaneous spatial and temporal extrapolation for video generation via incorporating the 3D-RoPE. On the VBench-I2V benchmark, VideoMAR achieves better performance compared to the Cosmos baseline, with much smaller model size, data scale, and GPU resources.

# 2 Related Work

## 2.1 Autoregressive Video Generation

**Raster-scan autoregressive models.** Similar to the VQ quantization and NTP paradigm in autoregressive image generation models [9, 24, 28, 21], some methods also employ this paradigm for autoregressive video generation [17, 30, 29, 25, 1]. For example, VideoPoet [17] employs a decoder-only transformer architecture to processes multi-modal inputs, incorporating a mixture of

multi-modal generative objectives. Cosmos [1] trains the autoregressive-based world foundation model via video generation.

**Mask-based autoregressive models.** Mask-based autoregressive models predict the masked tokens given the unmasked ones. They introduce a bidirectional transformer and predict randomly masked tokens by attending to unmasked conditions [3, 36, 11, 8, 39]. This paradigm enhances vanilla AR by predicting multiple tokens at every step. For example, MAGVIT [36] tackles various video synthesis tasks with a single model, via randomly masking the video sequence. Genie [3] proposes interactive video game generation in an unsupervised manner and generates videos frame-by-frame. Inspired by the continuous tokens in MAR [19], some recent works also propose to combine continuous tokens and masked generation models for video generation [8, 39]. For example, NOVA [8] generates the coarse features frame-by-frame and refines each frame with a spatial layer. MAGI [39] proposes complete teacher forcing by conditioning masked frames on complete observation frames.

## 2.2 Diffusion-based Long Video Generation

Recently, some diffusion-based video generation models attempt to extend the inference video length and achieve autoregressive-like video generation. These methods can be mainly divided into two types. The first type [31, 12, 34] basically follows video diffusion models to repetitively generate video clips, and sequentially cascade these video chunks to achieve longer video generation. For example, CausVid [34] transforms bidirectional models into fast autoregressive ones through distillation. The second type [5, 2, 14] applies varying levels of noise to different video chunks, imitating the causality of autoregressive generation. For example, DiffusionForcing [5] applies higher noise level to the later frame in each chunk and shifts across frames for the generation of more frames. MAGI-1 [2] follows this paradigm by applying varying noise levels to different chunks while maintaining the noise level identical for frames in each chunk. These methods still fall in the range of diffusion models, employing diffusion model for the generation of each frame or video chunk. Therefore, these methods are basically different from the autoregressive video generation, which employs AR models for the token-wise generation.

## 3 Preliminary

**Task definition and symbology setting.** This paper focuses on the image-to-video autoregressive video generation, which sequentially generates the next frame forming the complete video with the given initial image and text prompt. For notation clarity, we depict the symbology settings in Table 1.

**Autoregressive video generative model.** The limitations of NTP paradigm is clearly presented in the introduction part. In this section, we cast on the mask-based generation paradigm. Mask-based generation is a common paradigm for autoregressive image generation [4, 19], which randomly masks partial image tokens and predicts these masked tokens with remaining visible tokens. Compared to NTP, this paradigm has the advantage of parallel token generation in each step, significantly reducing the inference steps. When extended to autoregressive video generation, most existing methods [3, 36] treat the video sequence similarly as image, treating all tokens within each video frame equally and using bidirectional attention for the whole video.

Table 1: Symbology settings.

| | |
|---|---|
| $T \times H \times W$ | Frames $\times$ height $\times$ width in latent space. |
| $N/x_n$ | Number of video tokens / n-th token. |
| $T/S_t$ | Number of video frames / t-th frame. |
| $S_t^v/S_t^m$ | All visible / masked tokens in frame $t$. |
| $C$ | Text prompt condition. |

$$p\left(C, S_1, \ldots, S_T\right) = p\left(S_1^m, S_2^m, \ldots, S_T^m \mid C, S_1^v, S_2^v, \ldots, S_T^v\right). \tag{3.1}$$

However, such operation bears several limitations: **1)** These methods face the dilemma of either failing to inference frame-by-frame [36], or facing training-inference bias when performing temporally sequential inference[3]. For example, the previous context frames are partially masked during training. While, the frame-by-frame generation requires all the tokens in previous frames are available for the next frame prediction. **2)** It depends on fixed-length video frames, which can lead to poor scalability in context and issues with coherence over longer video durations. It sacrifices the unique context extension potential of the token-wise modeling in AR models, which is verified possible and important

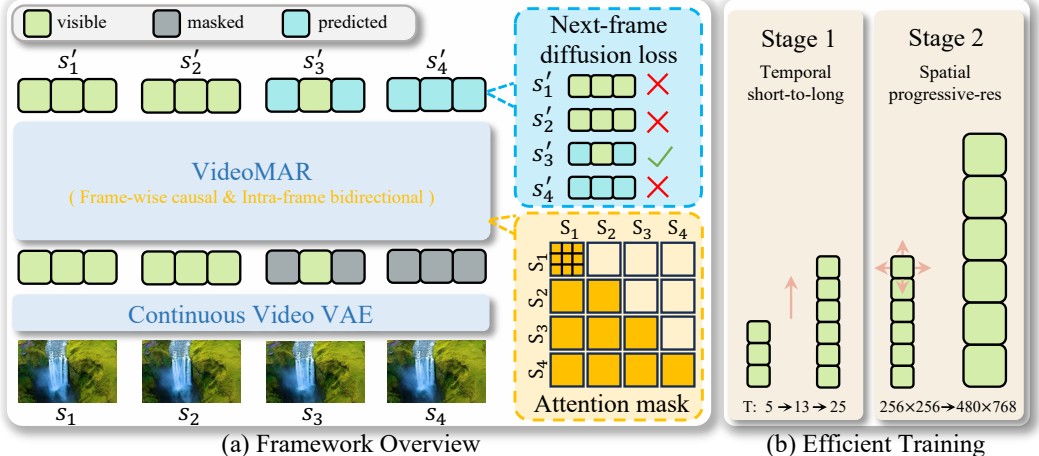

Figure 2: Framework of VideoMAR. *(a):* Training flowchart of VideoMAR. We employ frame-wise causal attention mask for temporal causality. Besides, we introduce the next-frame diffusion loss to the spatial partial-masked frame, which has complete previous frames. *(b):* Efficient training of VideoMAR. We apply temporal short-to-long curriculum learning and spatial progressive-resolution for reducing the training difficulty and cost of VideoMAR.

in language models [27]. **3)** It is incompatible with the inference optimization methods, such as KV cache. Recently, some methods attempt to optimize the mask-based generation [8, 39]. While, they either separate the temporal and spatial modeling, or double the sequence length. In this paper, starting from the desired way of sequentially generating each frame depending on all the previous context frames, we propose VideoMAR, a simple and novel paradigm for mask-based autoregressive video generation. VideoMAR is free from all these three limitations.

# 4 VideoMAR

## 4.1 Framework Overview

Video autoregressive model generates each frame conditioning on all the previous frames. Mask-based generation models generate the masked tokens with the remaining visible ones. Therefore, mask-based video AR model should generate the masked tokens at each frame with all the tokens in the preceding frames and the visible tokens in the current frame. To this end, we propose VideoMAR, the decoder-only mask-based autoregressive video generation model. As shown in Figure 2, VideoMAR works with continuous tokens via first compressing the video into continuous tokens with video VAE. Then, VideoMAR adopts frame-wise causal attention mask, enabling temporal causal and spatial bidirectional modeling.

**Next-frame loss.** To fit the temporal causal inference and mask generation properties of VideoMAR, we devise the next-frame diffusion loss as depicted in Figure 2. Specifically, we randomly mask partial tokens in certain frame $t \in [1, T]$. For the preceding frames before frame $t$, all the tokens are remain unchanged. For the frames after frame $t$, all the tokens are masked. This processed video sequence maintains temporal causality with frame-wise causal attention mask. The token-wise diffusion loss optimization is only applied to the masked tokens in frame $t$. The tokens after frame $t$ are free from loss optimization as not all tokens in the previous frames are available. During training, the frame $t$ is randomly chosen between $[1, T]$, spanning all the frames.

$$p(C, S_1, \ldots, S_T) = \prod_t^T p\left(S_t^m \mid C, S_1, \ldots, S_{t-1}, S_t^v\right). \tag{4.1}$$

## 4.2 Training

**Temporal short-to-long curriculum learning.** Due to the long sequence nature of video data, it would be quite difficult and inefficient to directly model this long sequence. Different from the joint

bidirectional video frames modeling of video diffusion models, video autoregressive model generates the next frame based on previous frames. This temporally sequential property highlights the priority of early frames for both training and inference.

To this end, we propose the temporal short-to-long curriculum training. Specifically, we first train on the short video clips with much shorter sequence length. In this stage, VideoMAR processes the capacity of clear visual quality and basic temporal motion modeling, with substantially reduced training cost and difficulty. When the early frames in the short clips are converged, we extend the video frame length to progressively capture larger temporal motion modeling capacity. This progressive short-to-long curriculum learning strategy successfully decompose the long sequence modeling difficulty to various phases in a very efficient way.

**Spatial two-stage progressive-resolution training.** In the first stage, VideoMAR trains on low resolution of $256 \times 256$. During this stage, we adopt the above mentioned short-to-long curriculum learning. With lower resolution and curriculum learning, VideoMAR possesses basic autoregressive video generation capacity with high efficiency. In the second stage, we finetune the model with higher resolution of $480 \times 768$ to support higher resolution video generation. To deal with the significantly increased token sequence, we adopt VAE with higher compression ratio. Thanks to the employed relative positional encoding, this finetuning stage is empirically verified efficient.

### 4.3 Inference

**Accumulation error.** AR models inherently suffer from exposure bias problem [38]. Specifically, each token is predicted with preceding GT tokens during training. However, during inference, all preceding tokens are the corresponding predicted ones, which may be incorrect. Such exposure bias is especially obvious for video generation which has long sequence length. To solve this, we first explore the importance and error of each generated frame. We find that the early-frame is vital for motion degree and suffer from relatively small accumulated error [18], while the late-frame focuses more on keeping visual quality and motion smoothness and suffers from more accumulated error, as shown in Figure 8. Furthermore, we uncover that temperature plays a key role in error suppression, where low temperature reduces the accumulated error (Detailed analyses and ablation of temperature on generation result is available in the supplementary material). With this observation, we propose the progressive temperature strategy, applying smaller temperature to the later frames. Specifically, the temperature varies from 1 to 0.9 across frames. The lower temperature in the later frame effectively suppresses the accumulated error and thus achieves much better visual quality. Besides, the slighter lower temperature is enough to keep motion smoothness given the previous dynamic video frames.

**Efficient inference.** VideoMAR generates video tokens via mask-based parallelized generation, while maintaining the frame-by-frame generation. This paradigm combines the advantage of the spatial token parallel generation and temporal KV cache acceleration. As shown in Table 2, we present the inference steps comparison with NTP paradigm, where VideoMAR significantly reduces the steps $20\times$ times (from 1440 to 64). Based on the spatial parallel generation, VideoMAR further reduces the inference time via enabling the KV cache (from 672s to 134s). Compared to NTP, our method achieves more than $10\times$ acceleration (from 1941s to 134s).

Table 2: **Efficient inference of VideoMAR**. VideoMAR combines the advantage of the spatial token parallel generation and temporal KV cache acceleration. Note that the NTP* is listed only for inference speed comparison, and we did not train such model.

| Methods | | Steps (spatial × temporal) | Inference time (s) |
|---|---|---|---|
| | NTP* | 1024x6 | 1310 |
| VideoMAR(stage1) | w/o KV Cache | 64x6 | 417 |
| | w/ KV Cache | 64x6 | 107 |
| | NTP* | 1440x6 | 1941 |
| VideoMAR(stage2) | w/o KV Cache | 64x6 | 671 |
| | w/ KV Cache | 64x6 | 134 |

**Spatial and temporal extrapolation ability.** Context token length extrapolation is a basic capacity of LLM, which can generate millions of tokens significantly longer than the training sequence length. Position encoding (PE) method plays a key role in this capacity. In this paper, we experiment with different PE methods, including the absolute cosine PE and RoPE [27] (extrapolation ability comparison between different PE methods are available in the supplementary material). In the

final implementation, we apply 3D-RoPE as the only position encoding method, and for the first time simultaneously boost the spatial and temporal extrapolation ability of video autoregressive model. VideoMAR can generate videos of varying resolutions and length, while only trained on fixed resolution and aspect ratio. Visual examples are available in Figure 4. We also depict more arbitrary scaling examples in Sec. D, where our method can flexibly generate videos of varying aspect ratios.

## 5 Experiments

### 5.1 Implementation Details

**Experimental setup.** We employ the general decoder-only architecture as the backbone of Video-MAR. The VideoMAR backbone consists of 36 transformer layers with a dimension of 1536. We mostly follow MAR [19] for the implementation of token-wise diffusion loss. The denoising MLP consists of 3 blocks with a dimension of 1280. We adopt the masking and diffusion schedulers from MAR [19], using a masking ratio between 0.7 and 1.0 during training, and progressively reducing it from 1.0 to 0 following a cosine schedule with 64 autoregressive steps during inference. In line with common practice [13], we train with a 1000-step noise schedule but default to 100 steps for inference. For the text prompt, following the practice in FAR [35], we employ Qwen2-1.5B [32] as our text encoder and adopt cross attention for text condition injection. For the visual tokenizer, we adopt Cosmos-Tokenizer [1]. For the first stage ($256 \times 256$ resolution), we employ Cosmos-Tokenizer with $4 \times 8 \times 8$ compression in the temporal and spatial dimensions. The temporal short-to-long curriculum learning is arranged with frame length order of (5, 13, 25). For the second stage ($480 \times 768$ resolution), we employ Cosmos-Tokenizer with $8 \times 16 \times 16$ compression. We utilize the AdamW optimizer [20] ($\beta_1 = 0.9$, $\beta_2 = 0.95$) with a weight decay of 0.02 and a base learning rate of $1e^{-4}$ in all experiments. All the weights are trained from scratch with 64 NVIDIA H20 GPUs.

**Datasets.** For image-to-video training, we employ 0.5M internal video-text pairs.

**Evaluation.** We use VBench-I2V [16] to evaluate the capacity of image-to-video generation across all the 9 dimensions. For a given text prompt, we randomly generate 5 samples, each with a video size of $25 \times 256 \times 256$ for the first stage and $49 \times 480 \times 768$ for the second stage. We employ classifier-free guidance with a value of 3.0 to enhance the quality of the generated videos in all evaluation experiments. Each latent frame is generated with 64 autoregressive steps

### 5.2 Quantitative Comparison

**VideoMAR is comparable with diffusion image-to-video models and significantly suppresses the AR counterpart with much lower training costs.** We perform a quantitative comparison with mainstream and cutting-edge image-to-video models, which can be divided into two types: diffusion models and autoregressive models. For diffusion model type, the compared models include I2VGen-XL [37], ConsistI2V [26], SEINE [7], VideoCrafter-I2V [6], CogVideoX-I2V [33], Step-Video-TI2V [15], and Magi-1 [2]. For the autoregressive type, Cosmos [1] is the most powerful autoregressive image-to-video model. As shown in Table 3, despite its significantly smaller size (1.4B vs. 5B&13B), data scale (0.5M vs. 100M), training cost (64 H20 GPUs vs. 10000 H100 GPUs), VideoMAR remarkably outperforms Cosmos (84.51 vs. 84.22) on the VBench-I2V benchmark across a variety of sub-dimensions. VideoMAR also rivals some diffusion-based image-to-video models including ConsistI2V and VideoCrafter-I2V with much lower training costs. For the latest SOTA diffusion models, like Step-Video-TI2V and Magi-1, our method still lags behind. While, our method focuses on the more challenging and promising AR-paradigm video generation and it's already quite convincing and promising of VideoMAR to achieve such performance with quite limited resources.

### 5.3 Qualitative Results

**High motion smoothness and visual quality.** We present the qualitative comparison in Figure 3. VideoMAR demonstrates high visual quality in each frame and smooth motions across adjacent frames. The motion type of VideoMAR contains both object motion, camera motion, and stable scene transitions. In contrast, Cosmos suffers from poor quality and details due to the employment of discrete tokens. Besides, the motions in Cosmos are mostly camera motion, with few object motions.

Table 3: **Image-to-video evaluation on VBench.** We have classified existing video generation methods into different categories for better clarity. The baseline data is sourced from VBench-I2V [16]. The data of Cosmos is tested with its official code and recommended parameters.

| Model | params | data | Total Score | I2V Score | Qual. Score | I2V Subj. | I2V Back. | Came. Moti. | Subj. Cons. | Back. Cons. | Moti. Smoo. | Dyna. Degr. | Aest. Qual. | Imag. Qual. |
|---|---|---|---|---|---|---|---|---|---|---|---|---|---|---|
| *Diffusion models* | | | | | | | | | | | | | | |
| Magi-1 | 24B | - | 89.28 | 96.12 | 82.44 | 98.39 | 99.00 | 50.85 | 93.96 | 96.74 | 98.68 | 68.21 | 64.74 | 69.71 |
| Step-Video-TI2V | 30B | 5M | 88.36 | 95.50 | 81.22 | 97.86 | 98.63 | 49.23 | 96.02 | 97.06 | 99.24 | 48.78 | 62.29 | 70.44 |
| CogVideoX-I2V | 5B | 35M | 86.70 | 94.79 | 78.61 | 97.19 | 96.74 | 67.68 | 94.34 | 96.42 | 98.40 | 33.17 | 61.87 | 70.01 |
| SEINE | - | 10M+ | 85.52 | 92.67 | 78.37 | 97.15 | 96.94 | 20.97 | 95.28 | 97.12 | 97.12 | 27.07 | 64.55 | 71.39 |
| I2VGen-XL | - | 35M | 85.28 | 92.11 | 78.44 | 96.48 | 96.83 | 18.48 | 94.18 | 97.09 | 98.34 | 26.10 | 64.82 | 69.14 |
| ConsistI2V | - | 10M | 84.07 | 91.91 | 76.22 | 95.82 | 95.95 | 33.92 | 95.27 | 98.28 | 97.38 | 18.62 | 59.00 | 66.92 |
| VideoCrafter-I2V | - | 10M | 82.57 | 86.31 | 78.84 | 91.17 | 91.31 | 33.60 | 97.86 | 98.79 | 98.00 | 22.60 | 60.78 | 71.68 |
| *Autoregressive models* | | | | | | | | | | | | | | |
| Cosmos | 5B | 100M | 84.16 | 92.51 | 75.81 | 95.99 | 97.36 | **25.56** | 97.12 | 96.59 | 99.47 | **20.33** | 55.82 | 59.90 |
| Cosmos | 13B | 100M | 84.22 | 92.60 | **75.83** | 96.17 | 97.35 | 25.43 | 97.69 | 96.77 | 99.40 | 18.70 | **55.84** | 60.15 |
| VideoMAR-stage1 | 1.4B | 0.2M | 82.56 | 91.74 | 73.39 | 96.64 | 96.24 | 16.80 | 97.08 | **98.78** | 99.32 | 13.01 | 52.05 | 51.61 |
| VideoMAR-stage2 | 1.4B | 0.5M | **84.82** | **94.02** | 75.61 | **97.85** | **98.38** | 21.62 | **97.13** | 97.20 | **99.57** | 10.98 | 55.81 | **62.34** |

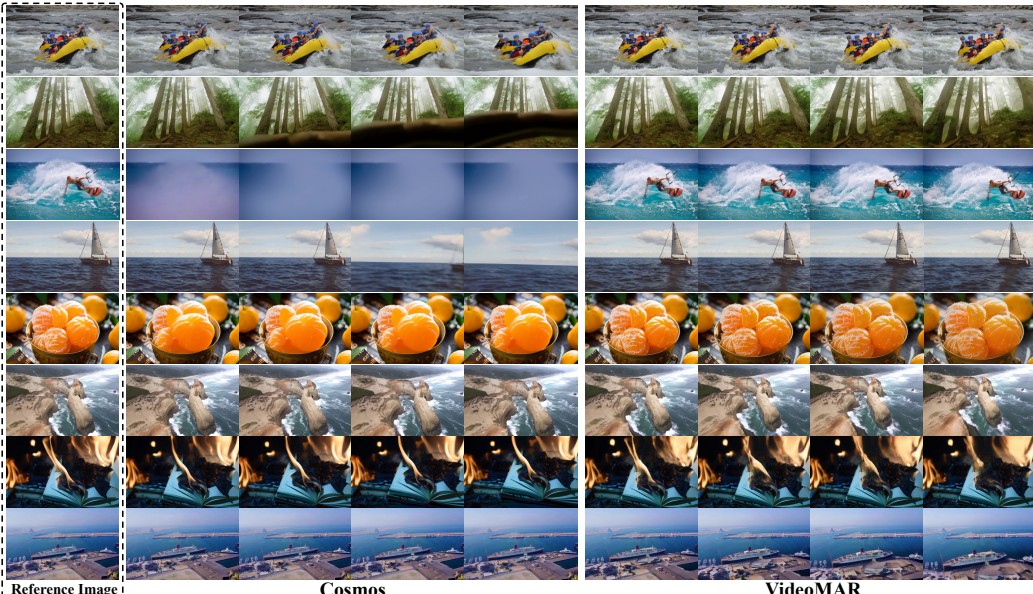

Reference Image     **Cosmos**     **VideoMAR**

Figure 3: Visual comparison between Cosmos 5B and VideoMAR on image-to-video generation. The motions in Cosmos are mostly camera motion, with few object motions. Cosmos is also prone to have failure cases, which induces abrupt object change and poor consistency. For example, the surfing man disappears in the third row, and the texture of oranges changes in the fifth row.

**Spatial and temporal extrapolation.** Context token extrapolation capacity is a desired property for autoregressive visual generation. For spatial dimension extrapolation, few current methods show such ability. For temporal extrapolation, some methods claim longer video generation ability, via repeatedly generating video chunks. In this paper, VideoMAR demonstrates the capacity of simultaneous spatial and temporal extrapolation to generate video of larger resolution and duration in Figure 4. This is achieved in a training-free manner without chunk-wise split.

**Extreme long video generation.** Besides the boosted resolution scaling ability, we further verify the extremely long video generation capacity of VideoMAR. As shown in Figure 5, we autoregressively generate videos of 20 second (12 fps), instead of repeatedly generating multiple video clips. While the maximum training length is 4 second. For the generation of such long video, we discuss three kv cache methods. **(1)** Full kv-cache. We cache all the history frames for the generation of the next frame. This strategy cache increasingly more features, inducing slower inference speed and larger memory consumption for the generation of latter frames. **(2)** Window kv-cache. This strategy maintains a constant length cache and is temporally smooth due to temporal locality. While, the

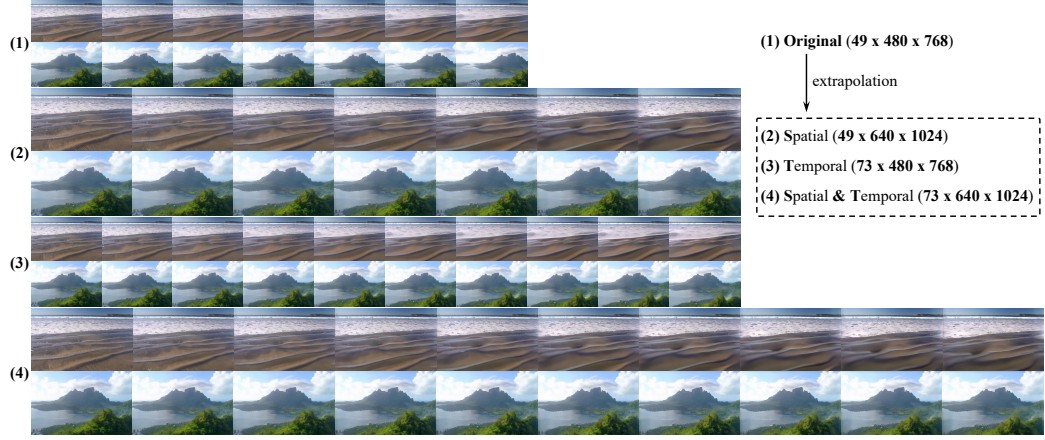

(1) Original (49 x 480 x 768)

extrapolation

(2) Spatial (49 x 640 x 1024)
(3) Temporal (73 x 480 x 768)
(4) Spatial & Temporal (73 x 640 x 1024)

Figure 4: Spatial and temporal extrapolation capacity of VideoMAR.

short memory also induces poor long-range consistency. **(3)** First frame & window kv-cache. The cached first frame provides the anchored spatial information and the window frames provide temporal smoothness. We empirically recommend the third strategy for its superior performance.

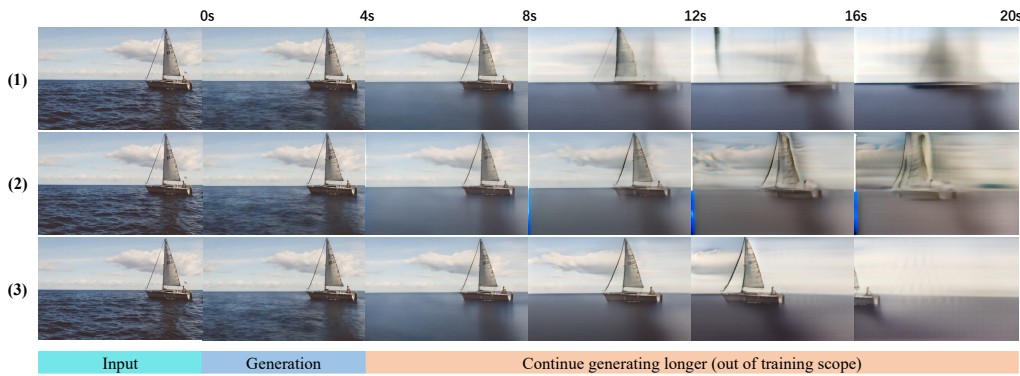

Figure 5: Extreme long video generation capacity of VideoMAR. VideoMAR can, for the first time, autoregressively generate longer videos far beyond the training length, instead of repeatedly generating multiple video clips. It supports $\times 3$ longer generation without severe degradation, and $\times 5$ longer generation with reasonable motions. Different rows represent various kv cache methods.

**Video-to-video generation.** Besides autoregressive image-to-video generation, VideoMAR also unlocks video-to-video generation. Given videos of various frame length, VideoMAR can generate more frames based on the given video. As shown in Figure 6, we present the visual results with two frames as condition. Video condition contains not only spatial prior but also temporal motion clues, which enables VideoMAR to have desired motion type and dynamic degree.

## 5.4 Ablations

In this section, we verify the effectiveness of each design of VideoMAR, including 1) the causal attention mask; 2) the next-frame diffusion loss; and 3) the temperature strategy. As shown in Table 4, we measure their effectiveness with the VBench-I2V metric. Obviously, each design is beneficial to the performance. Besides, the additional advantage of inference acceleration is analyzed in Table 2.

Table 4: Ablation study of VideoMAR in stage 1 on VBench-I2V.

| Frame Loss | Causal Attn | Temperature | Total Score | I2V Score | Quality Score |
|:---:|:---:|:---:|:---:|:---:|:---:|
| ✗ | ✗ | ✗ | 78.81 | 86.57 | 71.05 |
| ✓ | ✗ | ✗ | 79.76 | 87.66 | 71.86 |
| ✓ | ✓ | ✗ | 80.72 | 88.90 | 72.54 |
| ✓ | ✓ | ✓ | 82.56 | 91.74 | 73.39 |

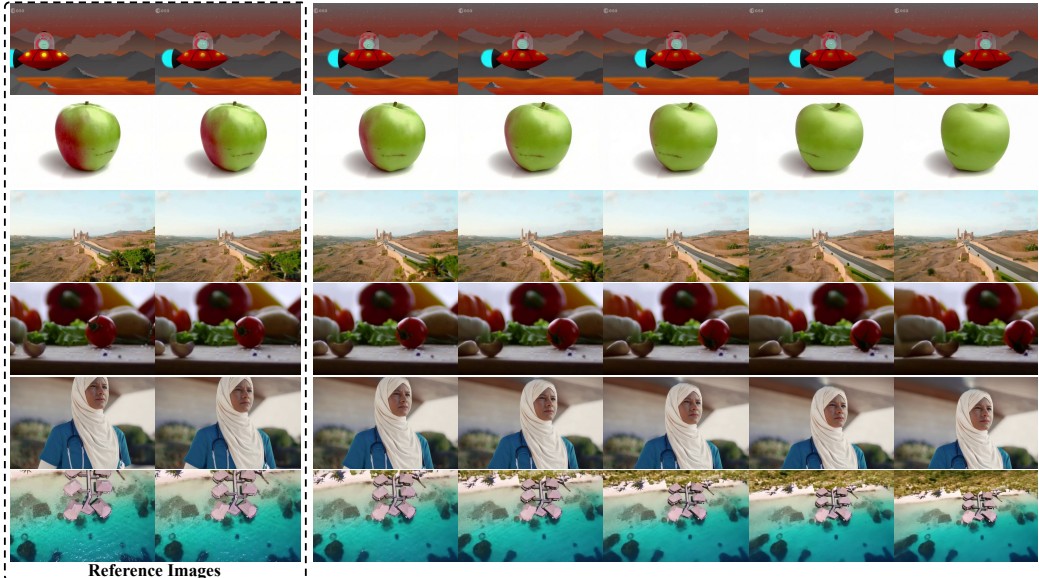

Figure 6: Visual results of VideoMAR on video-to-video generation, with two frames condition.

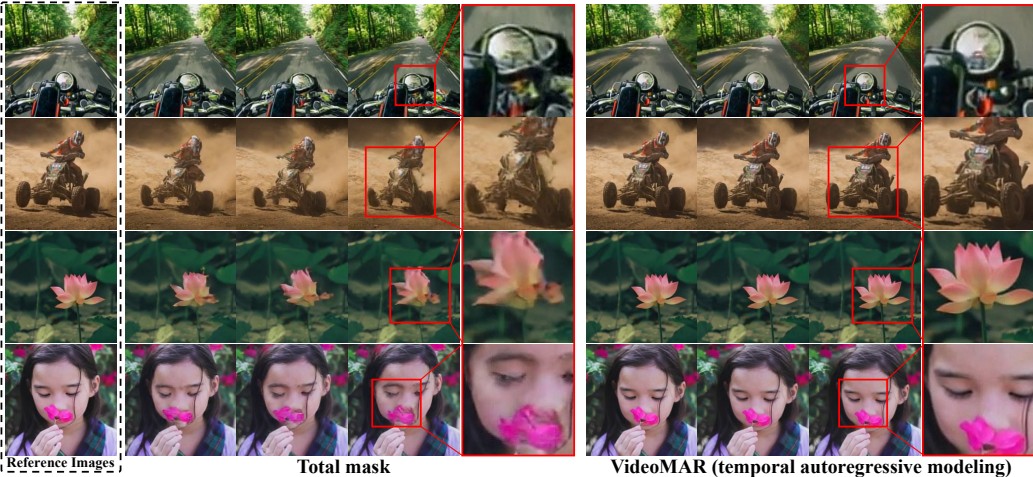

Figure 7: Visual comparison between the total mask baseline and our temporal autoregressive modeling VideoMAR in stage one. The baseline has poor local consistency with the reference image.

**Effectiveness of temporal autoregressive modeling.** To highlight the advantages of temporal autoregressive modeling, we denote the baseline as *Total mask* (w/o Frame Loss, w/o Causal Attn, w/ Temperature strategy), which treats the video sequence similarly as images, as shown in Equation 3.1. As shown in Figure 7, temporal autoregressive modeling obviously has higher visual quality and smoother motion than the baseline. The fully random masked generation of the baseline method is prone to have poor consistency with the reference image in local regions, *e.g.,* the motorcycle in the second row is broken, and the human face in the last row is distorted.

**Temperature strategy & Accumulation error.** To validate the effectiveness of temperature strategy for reducing accumulation error, we present the visual comparison in Figure 8. Without the temperature strategy, the late frames of the generated video tend to collapse due to accumulated error in the previous frames. For example, the buildings suffer from severe distortion and lose its structural shape. In contrast, VideoMAR adopts low temperature in the late frames, which maintains the dynamic degree across frames and high visual quality of each frame.

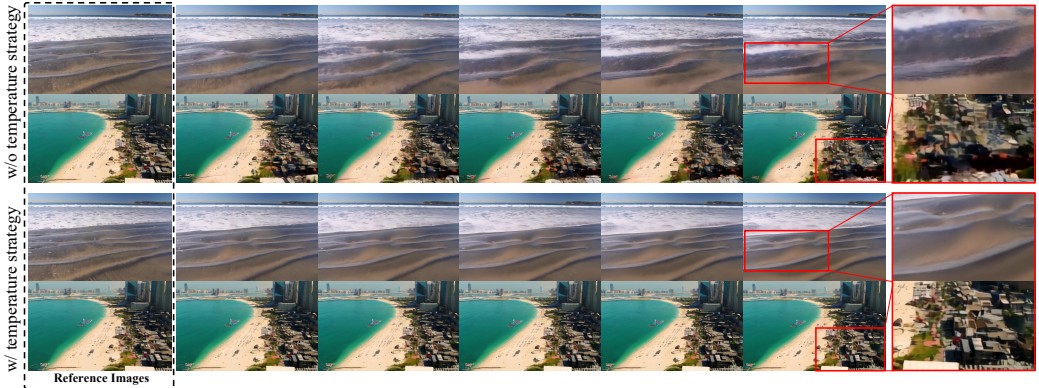

Figure 8: The effects of the temperature strategy on visual results. The late frames are prone to suffer from large accumulated errors and poor quality without the incorporation of the temperature strategy.

## 6   Discussion and Future Work

Despite the superior results achieved, we admit that there still exist some limitations that are worth further exploration: **1)** VideoMAR has the capacity of multiple tasks unification within this single paradigm, including text-to-image, text-to-video, image-to-video, video-to-video, and video editing. However, due to limited resources, we first explore image-to-video and video-to-video tasks, and leave other tasks in future work. **2)** VideoMAR can naturally function as interactive world model via replacing the prompt with frame-level interactive action condition. We will verify this in future work.

## 7   Conclusion

In this paper, we propose VideoMAR, a concise and efficient decoder-only autoregressive image-to-video generation model with continuous tokens, composing temporal frame-by-frame and spatial masked generation. VideoMAR retains complete previous context frames and introduces the frame-wise next-frame loss for training. Besides, we solve the huge cost and difficulty of long sequence autoregressive modeling with several crucial designs in both training and inference. On the VBench-I2V benchmark, VideoMAR achieves cutting-edge performance.

## 8   Acknowledgment

This work was supported by the Anhui Provincial Natural Science Foundation under Grant 2108085UD12. We acknowledge the support of GPU cluster built by MCC Lab of Information Science and Technology Institution, USTC. The AI-driven experiments, simulations and model training were performed on the robotic AI-Scientist platform of Chinese Academy of Sciences.

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

# A  Position Encoding

In this paper, we mainly verify two types of position encoding methods: 1) absolute cosine position encoding, and 2) relative position encoding.

**Absolute cosine position encoding.** Absolute positional encoding assigns a unique encoding vector to each position in the sequence, which is usually applied to fixed-length sequence. However, it neglects relative relationships and suffer from poor extrapolation of length. Unseen sequence lengths during training usually lead to performance degradation. As shown in upper part of Figure 9, the extrapolated frames fail to maintain the consistency.

**Relative position encoding.** Relative position encoding (RoPE) is the dominant position encoding method in language models, demonstrating great success in sequence length extrapolation. For video data, we adopt the 3D-RoPE, covering both the spatial and temporal dimensions. As shown in bottom part of Figure 9, RoPE endows our method with superior length extrapolation ability, with negligible visual quality degradation.

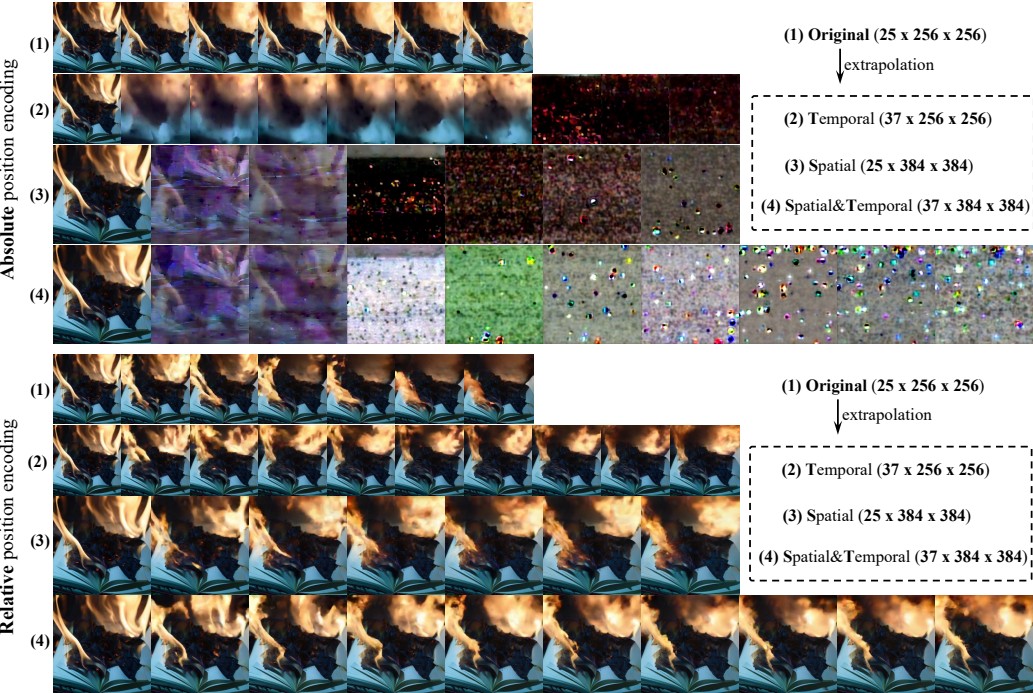

Figure 9: Visual comparison between absolute and relative position encoding methods.

# B  More Ablations on Temperature Strategy

In this section, we present more analyses of the temperature. As demonstrated in the main paper that temperature plays a key role in accumulation error suppression. In Table 5, we demonstrate the VBench-I2V scores under different temperatures. It is obvious that reducing the temperature helps consistently elevates the performance, e.g., the I2V Subject Score, the Subject Consistency Score, the Motion Smoothness Score, the Aesthetic Quality Score, etc.

Note that reducing the temperature may also negatively affect the dynamic degree score according to the table. While, in practical experiments, we find that the dynamic degree change is actually much small across various temperatures. The larger dynamic degrees in the bottom row are mainly attributed to the existence of failure cases, which can have unreasonable generation in late frames due to accumulation error.

To this end, as demonstrated in the main paper, this paper adopts the progressive temperature strategy to allocate lower temperature for the later frames. Specifically, we adopt the exponential function $0.9+10^{-(t+1)}$, with the frame $t \in [0, T]$. Note that employing a constant and relative low temperature

(e.g., 0.9) can achieve similar performance, but with slightly lower Dynamic Degree score. In practical usage, these two setting are both recommended.

Table 5: Ablations on the temperature strategy.

| Temperature | Total Score | I2V Subj. | I2V Back. | Subj. Cons. | Back. Cons. | Moti. Smoo. | Dyna. Degr. | Aest. Qual. | Imag. Qual. |
|---|---|---|---|---|---|---|---|---|---|
| 1.00 | 82.19 | 95.22 | 97.71 | 89.70 | 95.59 | 98.95 | 24.80 | 50.94 | 55.18 |
| 0.98 | 83.13 | 96.06 | 98.08 | 91.77 | 96.14 | 99.21 | 23.58 | 52.38 | 56.71 |
| 0.96 | 83.56 | 96.49 | 98.21 | 92.92 | 96.36 | 99.34 | 20.73 | 53.17 | 58.11 |
| 0.94 | 84.02 | 96.94 | 98.26 | 93.92 | 96.80 | 99.43 | 19.31 | 53.72 | 59.00 |
| 0.92 | 84.31 | 97.41 | 98.33 | 95.06 | 96.94 | 99.49 | 16.67 | 54.34 | 60.18 |
| 0.90 | 84.82 | 97.92 | 98.39 | 97.17 | 97.27 | 99.58 | 9.87 | 55.95 | 62.48 |
| Ours | 84.82 | 97.85 | 98.38 | 97.13 | 97.20 | 99.57 | 10.98 | 55.81 | 62.34 |

## C  Prompts for Videos in Main Manuscript

Due to page limit, we omit the corresponding prompts for each video in the main manuscript. In this section, we list the prompts of each Figure.

Prompts of Figure 1 in the main manuscript:

- tangerines in a metal bowl on a table
- Close up of eye blink
- Small fireworks explode and unfold in the air
- Tall trees with thick trunks and green foliage dominate the scene, set in a misty forest with a dirt path winding through the undergrowth. The atmosphere is foggy, creating a mystical and serene ambiance. The ground is covered with leaves and small plants, and the light filtering through the trees is soft and diffused. The overall style is realistic, with a slight blur due to the mist.
- Flower bloom

Prompts of Figure 3 in the main manuscript:

- A group of people in a yellow raft is rowing through turbulent waters
- Tall trees with thick trunks and green foliage dominate the scene, set in a misty forest with a dirt path winding through the undergrowth. The atmosphere is foggy, creating a mystical and serene ambiance. The ground is covered with leaves and small plants, and the light filtering through the trees is soft and diffused. The overall style is realistic, with a slight blur due to the mist.
- a man on a surfboard riding a wave in the ocean
- a sailboat is drifting on the ocean
- tangerines in a metal bowl on a table
- Aerial view of a coastal landscape featuring large, rugged rock formations jutting out into the ocean. The rocks are light brown with a rough texture, surrounded by blue-green waves crashing against them. The coastline is sandy with patches of green vegetation. The background includes a hazy sky and distant landforms. The overall style is realistic with clear, detailed imagery.
- a book on fire with flames coming out of it
- A large cruise ship with a red funnel is docked at a port, surrounded by several smaller boats and industrial structures. The port area includes warehouses, parking lots, and various equipment. The scene is set against a backdrop of calm blue water, with a long pier extending into the distance. The overall style is realistic, with clear visibility and no noticeable blur. The background includes distant landmasses and a clear sky.

Prompts of Figure 4 in the main manuscript:

- Waves gently wash over a sandy beach, creating intricate patterns in the wet sand. The water is clear and slightly foamy as it moves in and out. The horizon is visible in the distance with a clear blue sky above. The beach is empty, and the scene is calm and serene. The overall style is realistic with high clarity and no noticeable blur.

- A lush, green mountainous landscape with dense vegetation overlooks a serene body of water. The central mountain has a rugged, steep peak, and the surrounding area features rolling hills and valleys. The sky is partly cloudy with patches of blue, casting soft shadows on the terrain. The scene is bright and clear, with no visible human presence. The overall style is realistic, capturing the natural beauty and tranquility of the environment.

Prompts of Figure 5 in the main manuscript:

- A small, red spaceship with a transparent dome is floating in a vast, mountainous landscape. The spaceship has a sleek, rounded design with a glowing blue thruster at its rear. Inside the dome, a humanoid figure with a green complexion and a round head is seated, facing forward. The figure appears to be wearing a spacesuit, though the details of the suit are not clearly visible.The spaceship is the main object, and it is consistently centered in the scene. The background features a series of dark, jagged mountains against a reddish-orange sky, suggesting a desolate, alien environment. The landscape is barren, with no vegetation or structures visible, emphasizing the isolation of the scene. The spaceship remains at a relatively consistent distance from the mountains, maintaining a mid-air position above the ground.Throughout the sequence, the spaceship moves steadily from left to right across the screen. The shot is a medium shot, capturing the entire spaceship and part of the surrounding environment. The style of the scene is animated, with a somewhat cartoonish and colorful aesthetic, reminiscent of a Pixar-like animation. The humanoid figure inside the spaceship appears calm and focused, with no discernible facial expressions due to the distance and the dome's transparency.

- The video features a close-up of a green apple with a gradient from green to red, showcasing its textured skin and a small yellow spot. As time passes, the apple's glossy surface reflects light, emphasizing its freshness and natural beauty. The fruit's stem is brown, and it has a small yellow spot near the stem, which may indicate maturity or a natural imperfection. The apple's vibrant colors and the subtle shadows beneath it highlight its three-dimensional form and the detailed texture of its skin.

- The video begins with a wide shot of a large, ornate building with two prominent towers, situated in a vast, open landscape. The building is surrounded by dry, barren land with patches of greenery. A road runs parallel to the building, with a few cars visible on it. The sky is clear with a few clouds, and the overall color palette is dominated by earthy tones. As the video progresses, the camera moves closer to the building, providing a clearer view of its architectural details. The building appears to be made of stone or similar material, and its design suggests it may be of historical or cultural significance. The surrounding landscape is mostly flat, with a few hills in the distance. The video ends with a closer shot of the building, highlighting its grandeur and the serene environment in which it is located.

- The video features a series of close-up shots of fresh vegetables on a wooden cutting board, with a focus on cherry tomatoes, carrots, bell peppers, lettuce, and garlic. The vegetables are arranged with coarse salt and rosemary, suggesting preparation for a flavorful dish. The lighting is soft, highlighting the textures and colors of the produce, creating a warm, inviting atmosphere. As the frames progress, the arrangement of the vegetables slightly changes, with the addition of a yellow bell pepper and a purple onion, and the garlic cloves are shown in various states, from whole to peeled. The consistent theme is the anticipation of cooking with fresh ingredients.

- A female doctor, dressed in blue scrubs and a white hijab, stands confidently outdoors, her hands in her pockets. She wears a stethoscope around her neck, indicating her medical profession. The setting is a modern healthcare facility with a blurred background, suggesting a professional environment. Over time, her expression remains serious and focused, reflecting her dedication to her role. The lighting suggests it could be early morning or late afternoon. The background includes greenery and a building, hinting at a tranquil yet

professional atmosphere. Her attire and demeanor convey a sense of professionalism and cultural modesty.

- Aerial view of a tropical resort featuring several overwater bungalows with thatched roofs arranged in a symmetrical pattern extending from a central pier. The bungalows are situated in clear, turquoise water with visible darker patches indicating underwater features. The scene transitions from a closer view of the bungalows to a wider view, revealing a sandy beach lined with lush green vegetation and additional resort structures. The overall style is realistic, capturing the serene and picturesque nature of the tropical location.

Prompts of Figure 6 in the main manuscript:

- a motorcycle driving down a road
- a person riding an atv on a dirt track
- a pink lotus flower in the middle of a pond
- a young girl smelling a pink flower

Prompts of Figure 7 in the main manuscript:

- Waves gently wash over a sandy beach, creating intricate patterns in the wet sand. The water is clear and slightly foamy as it moves in and out. The horizon is visible in the distance with a clear blue sky above. The beach is empty, and the scene is calm and serene. The overall style is realistic with high clarity and no noticeable blur.
- A coastal cityscape with numerous tall, modern skyscrapers in light brown and beige hues lines the background, transitioning to a sandy beach with a turquoise sea in the foreground. The beach is populated with sunbathers and beachgoers, with several colorful umbrellas and small boats visible on the water. The scene shifts from a view of the buildings to a broader perspective of the coastline, revealing more of the beach and the sea. The water is clear and calm, reflecting the sunlight. The overall style is realistic, with high clarity and vibrant colors.

## D   Arbitrary Scaling

In the main manuscript, we mainly demonstrate the spatial and temporal extrapolation capacity of our method. In this section, we show that our method can perform arbitrary resolution scaling, including temporal longer/shorter, spatial larger/smaller generation and their arbitrary combination. To show such capacity, we present four more resolution ($H \times W \times T$) examples, including 1) $240 \times 384 \times 25$ in Figure 10, 2) $240 \times 384 \times 73$ in Figure 11, 3) $240 \times 768 \times 49$ in Figure 12, and 4) $480 \times 768 \times 37$ in Figure 13. Note that the model is trained on $480 \times 768 \times 49$.

## E   More Visual Comparisons with the Baseline Method

We present more visual comparisons between the Cosmos baseline and our method on image-to-video generation in Figure 14 and 15.

## F   More Visual Results

We present more visual results of our method on image-to-video generation in Figure 16, 17, and 18. We present more visual results of our method on video-to-video generation in Figure 19, 20, and 21.

## G   Inference Time Comparison with Baseline Method

We compare the inference time between the Cosmos baseline and our method in Table 6. Our method achieves comparable inference time than the Cosmos baseline. While, the efficiency of VideoMAR can be further improved with smaller autoregressive steps and diffusion steps. We currently employ autoregressive steps of 64 and diffusion steps of 100.

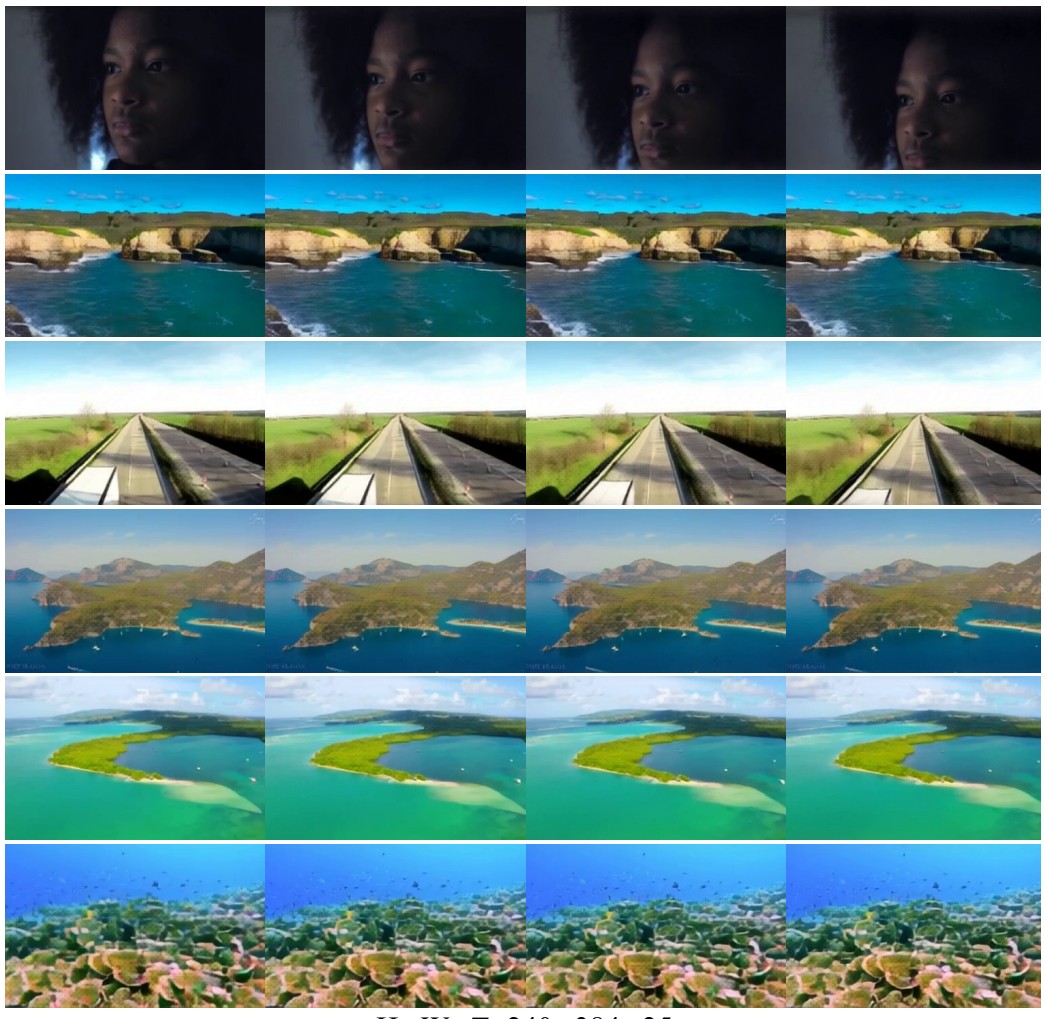

H×W×T=240×384×25

Figure 10: Arbitrary resolution scaling video samples with $H \times W \times T = 240 \times 384 \times 25$.

Table 6: Inference time comparison between Cosmos and VideoMAR.

| Methods | Cosmos-5B | Cosmos-13B | VideoMAR-stage1 | VideoMAR-stage2 |
|---------|-----------|------------|-----------------|-----------------|
| Time/s | 93 | 207 | 107 | 134 |

## H  Analyses of Motion Degree

Our method processes basic motion type in a very efficient way with limited resources. While, it still lacks in large and complex motions, due to small data scale and from-scratch training. In this section, we further verify that our method can generate complex motions, once given such data. Specifically, we collect two motion types, covering 1) blink and 2) blossom, each with several video samples. We finetune our model on these collected data respectively. In Figure 22, Our method can successfully replicate these complex and large motions. It is thus a future work to enlarge training data scale with more complex motions to improve the generalization ability of our method.

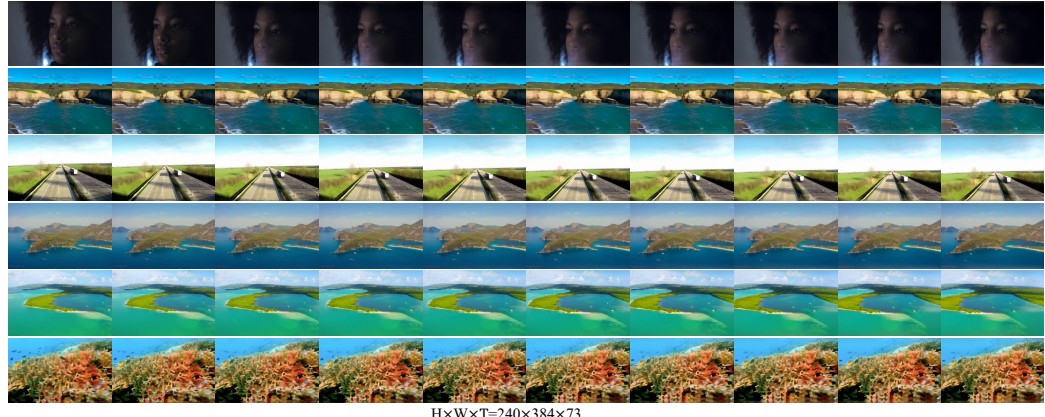

H×W×T=240×384×73

Figure 11: Arbitrary resolution scaling video samples with $H \times W \times T = 240 \times 384 \times 73$.

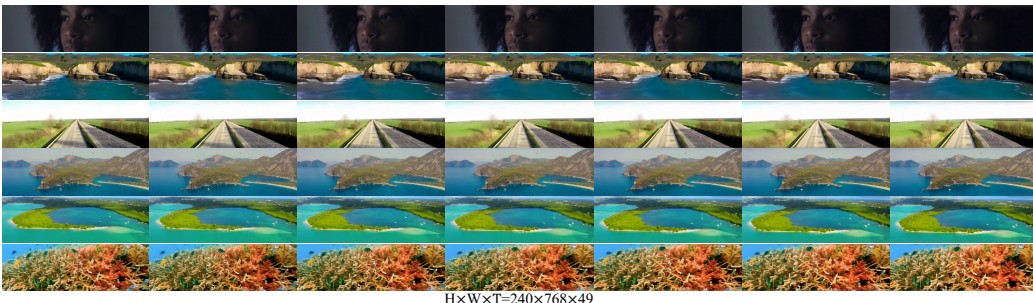

H×W×T=240×768×49

Figure 12: Arbitrary resolution scaling video samples with $H \times W \times T = 240 \times 768 \times 49$.

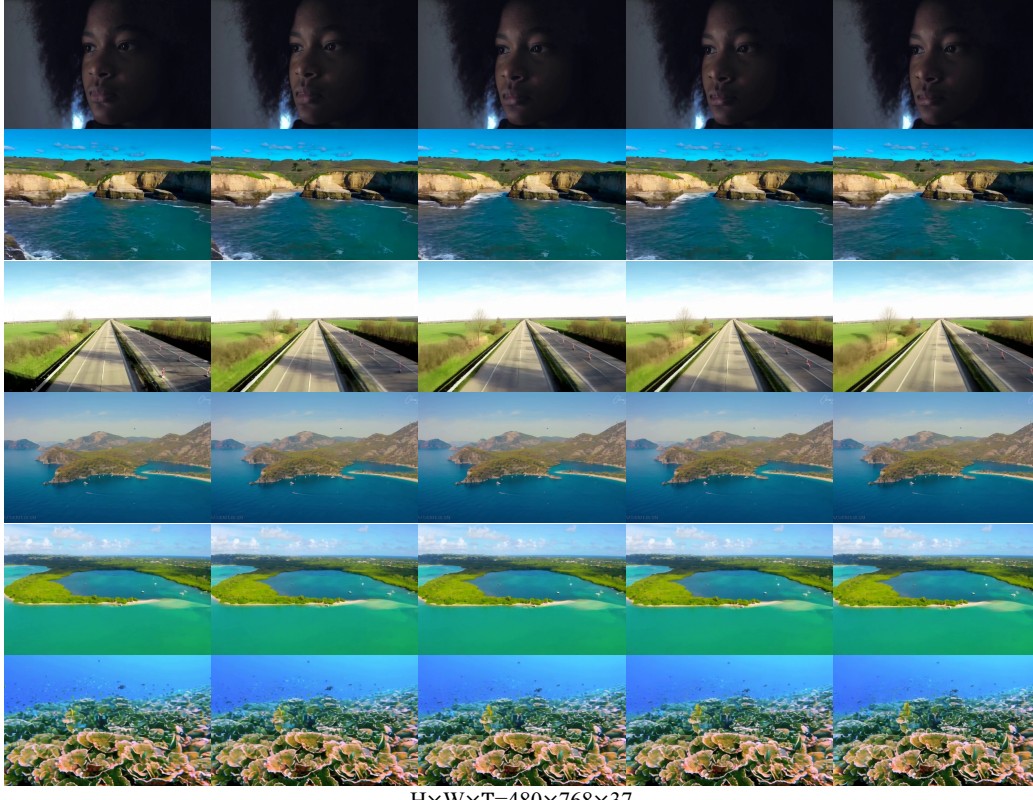

H×W×T=480×768×37

Figure 13: Arbitrary resolution scaling video samples with $H \times W \times T = 480 \times 768 \times 37$.

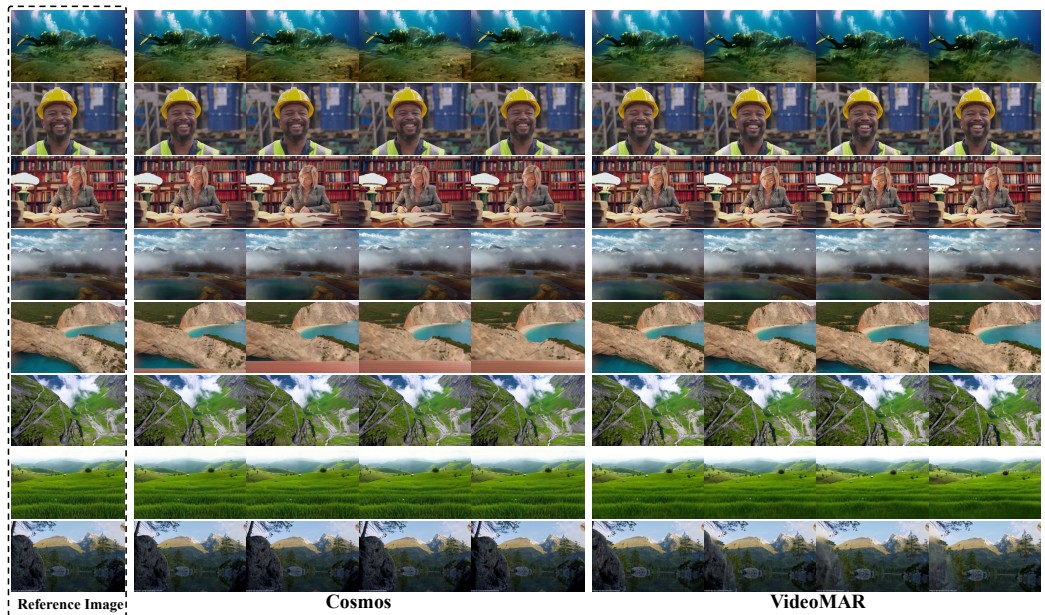

Figure 14: Additional visual comparison between Cosmos and our method on image-to-video generation.

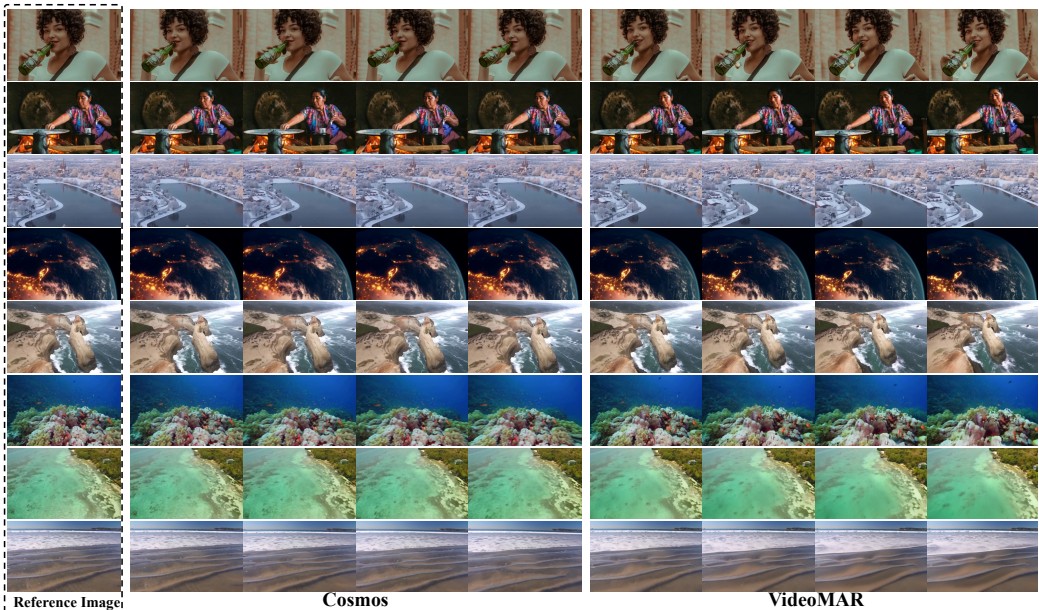

Figure 15: Additional visual comparison between Cosmos and our method on image-to-video generation.

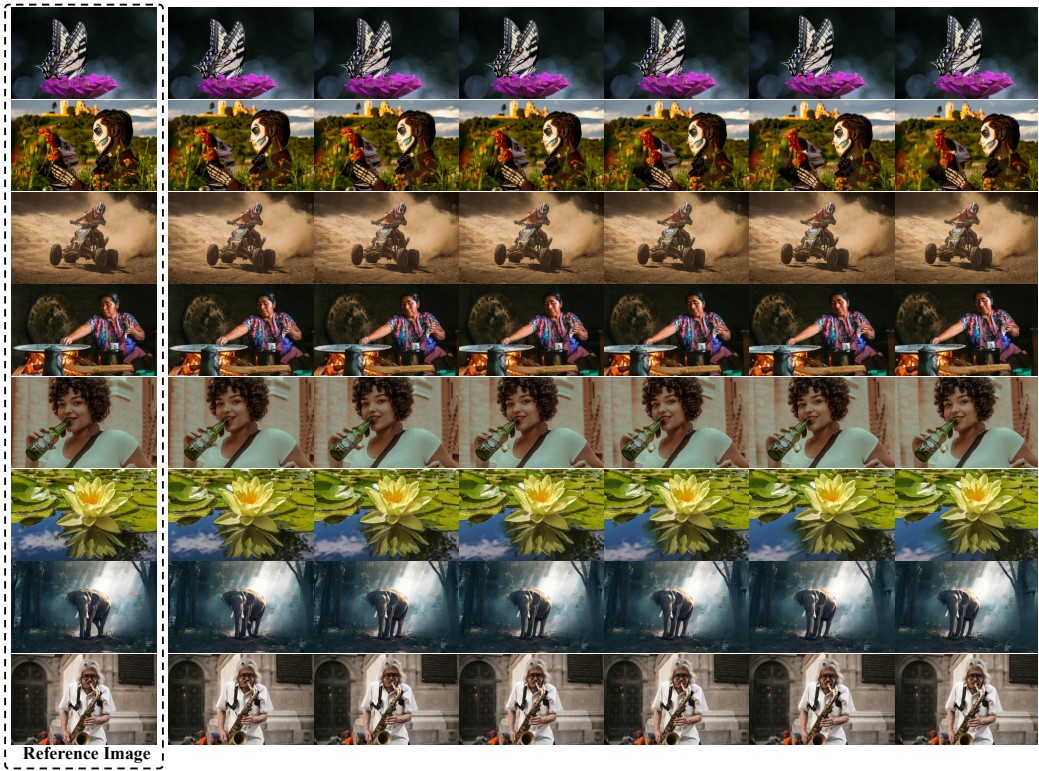

Figure 16: Additional visual results of our method on image-to-video generation.

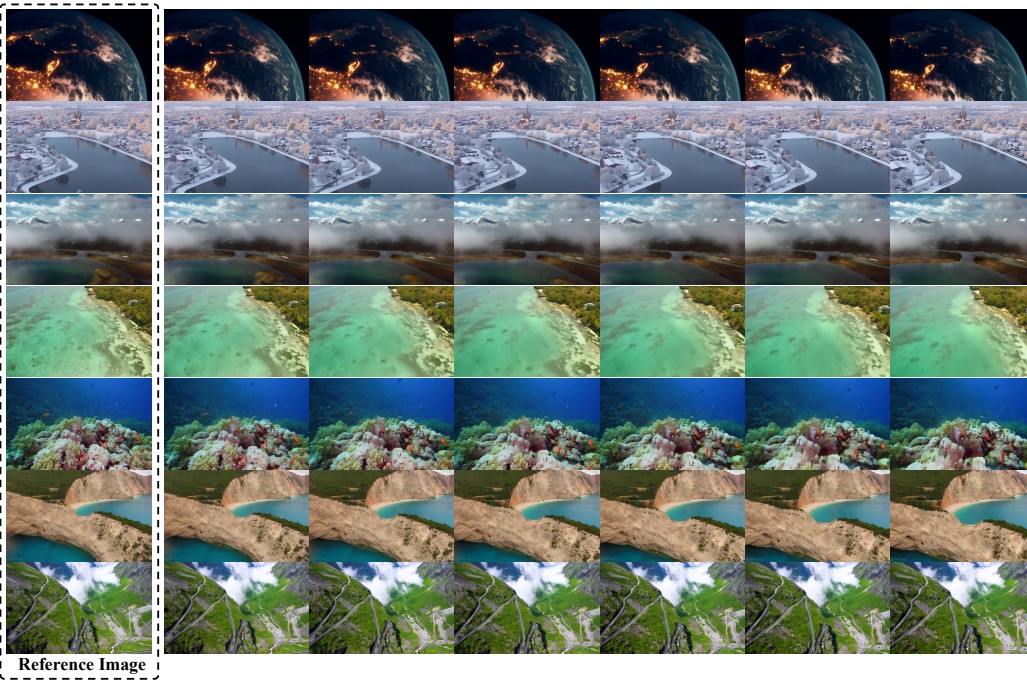

Figure 17: Additional visual results of our method on image-to-video generation.

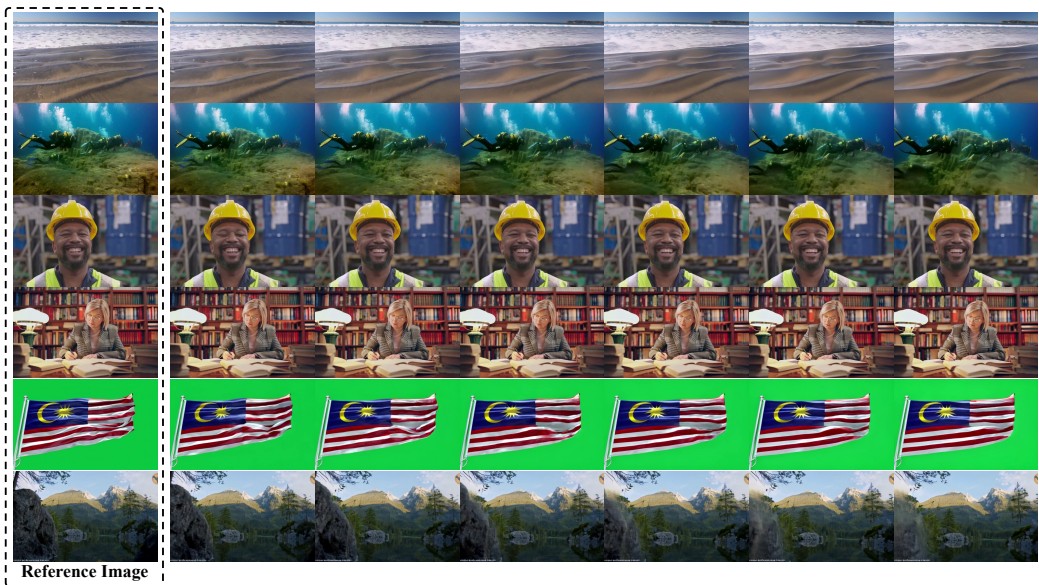

Figure 18: Additional visual results of our method on image-to-video generation.

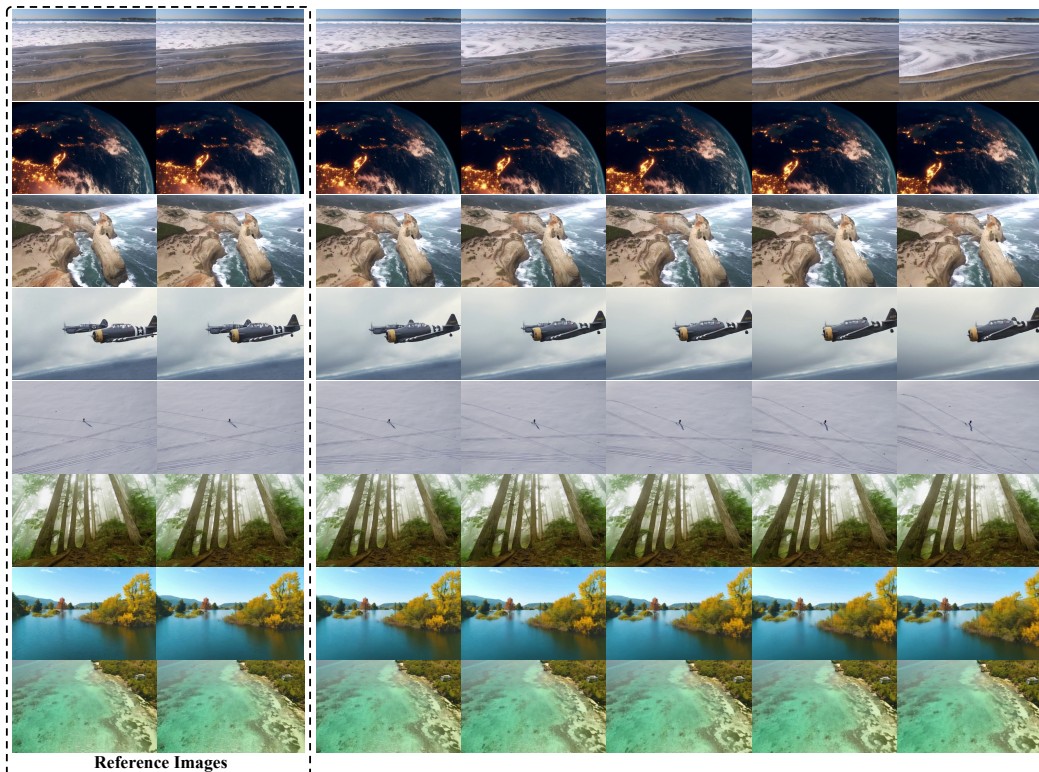

Figure 19: Additional visual results of our method on video-to-video generation.

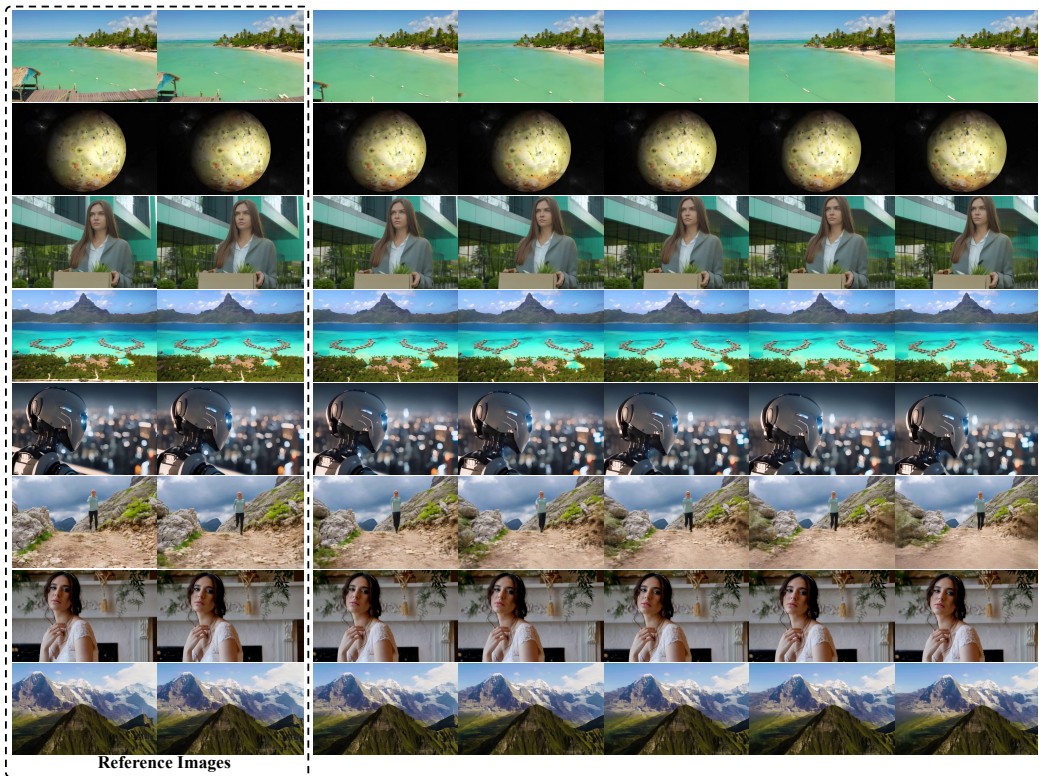

Reference Images

Figure 20: Additional visual results of our method on video-to-video generation.

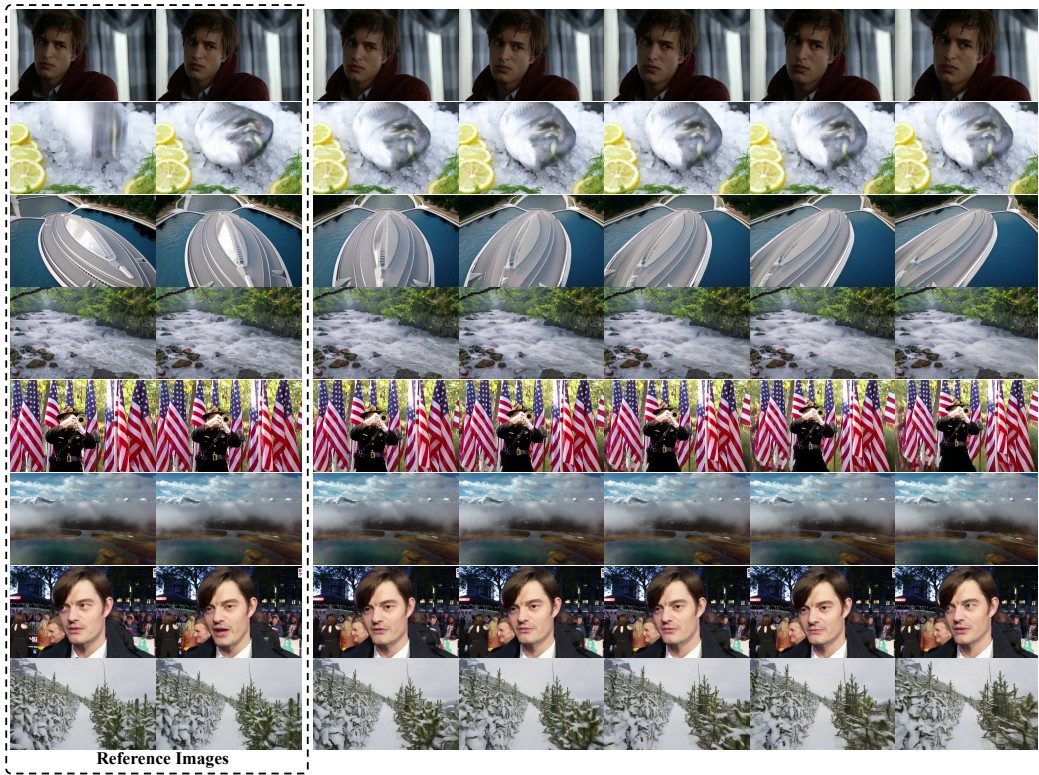

Reference Images

Figure 21: Additional visual results of our method on video-to-video generation.

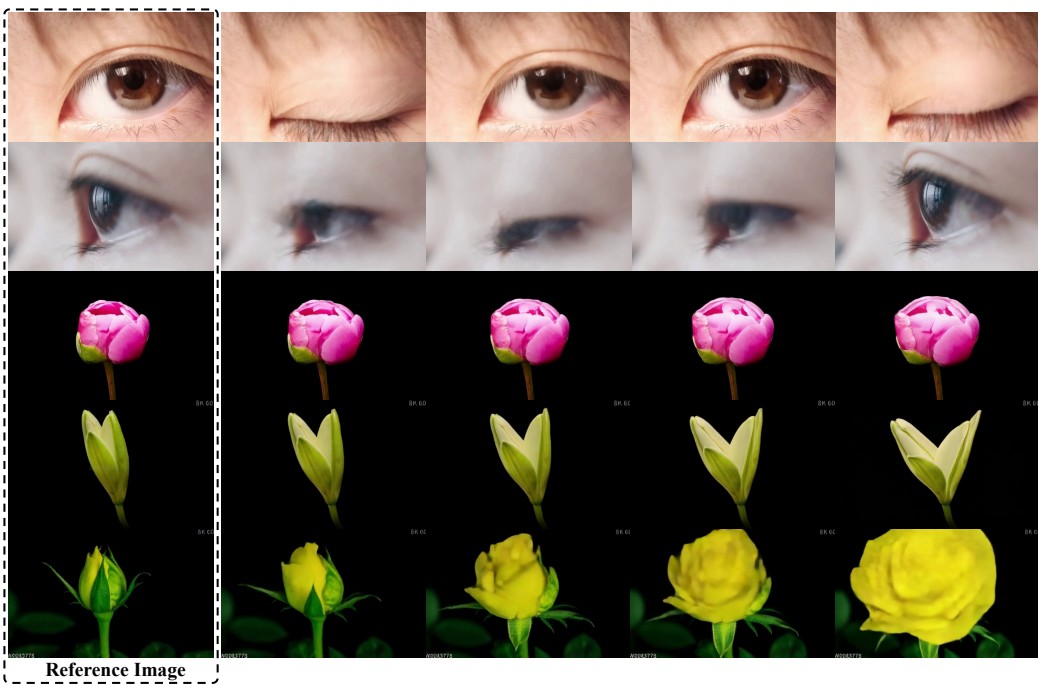

**Reference Image**

Figure 22: Finetuning results on the two additional collected motion types (blink and blossom) of our method on image-to-video generation.

