# OpenReview forum: "VideoMAR: Autoregressive Video Generation with Continuous Tokens"
_NeurIPS.cc/2025/Conference — NeurIPS 2025 poster_

### Official Review · Reviewer_Xwue · 2025-06-29

**Clarity:** 2
**Significance:** 2
**Originality:** 2
**Rating:** 4
**Confidence:** 4

**Summary:**

This paper presents VideoMAR, a masked autoregressive (MAR) framework for video generation. VideoMAR applies a causal masked autoregressive formulation for temporal modelling, while using bidirectional attention for spatial modelling. This separation enables KV caching, significantly improving inference efficiency compared to standard next-frame prediction models. In addition, the authors employ progressive training strategies (short-to-long and low-res-to-high-res) to enable 768p resolution video generation. Evaluated on VBench, VideoMAR outperforms Cosmos (another autoregressive baseline) while remaining less competitive than recent diffusion-based models that require much more compute.

**Questions:**

See weakenesses.

**Ethical Concerns:**

["NO or VERY MINOR ethics concerns only"]

**Final Justification:**

The rebuttal has resolved most of my concerns.

**Limitations:**

No. Would be great to add a small limitations section to discuss the model's current fallback as well as different areas to improve.

**Paper Formatting Concerns:**

No.

**Quality:**

2

**Strengths And Weaknesses:**

The paper proposes a clean, well-motivated design. The use of KV caching provides a substantial inference speed-up, and the generation quality over Cosmos demonstrates a solid performance improvement.

[Weaknesses and Suggestions]
- The core design of VideoMAR appears to be a direct extension of masked autoregressive (MAR) frameworks originally developed for image modeling. Similar video modelling designs using MAR, as used here, have already been explored in prior works such as NOVA, MAGI, and MarDini. The progressive training techniques for longer videos and higher resolution are widely adopted in the literature, especially in diffusion-based models, and also present in the aforementioned autoregressive works. The discussion in Lines 109–121 attempts to contrast VideoMAR with non-MAR video models, but fails to distinguish VideoMAR meaningfully from existing MAR-based models. Specifically, point 3 seems to critique the separation of temporal and spatial modeling, yet this is often considered a strength (e.g., in MarDini), as it enables more scalable architectures in a even more parameter-efficient way. More clarification is needed on why this is framed as a limitation in existing work and how VideoMAR differs.
- In table 3, the categorisation of models into “Diffusion” and “Autoregressive” is somewhat inconsistent and confusing:
    - The proposed model still relies on a diffusion framework, so calling it purely autoregressive may be misleading.
    - Models such as MAGI-1, which appear in the “Diffusion” category, are actually autoregressive in design.
    - Cosmos, the main baseline, is not a pure video generation model—it is more of a pretrained system for robotics and control tasks, which makes it an imperfect point of comparison. Including more relevant autoregressive video generation baselines would strengthen the empirical evaluation.
-  From the appendix samples, the generated videos appear to exhibit low frame rates and minimal object motion. It would be helpful for the paper to 1. Explicitly report the frame rate (FPS) of generated videos. 2. Include training and generation configurations that impact motion dynamics (e.g., frame stride, temporal length, augmentation strategy).

- The term “video-to-video” used in the paper may be misleading. The setup described, generating future frames conditioned on two reference frames, is better described as video extension. Typically, “video-to-video” refers to style transfer, semantic layout preservation, or conditioned transformation between videos.

---

> ### Author Rebuttal · Authors · 2025-07-30
>
> **Q1: VideoMAR appears to be a direct extension of masked autoregressive (MAR) frameworks.**
>
> We respectfully disagree with this assessment. VideoMAR represents a significant advancement over standard MAR frameworks by specifically tailoring the masking mechanism to the unique properties of video.
>
> **First**, our core contribution lies in the exploration and integration of video-specific masking strategies. We systematically investigated multiple approaches, including a total mask strategy that randomly masks entire frames, to find the optimal design. This thorough investigation goes far beyond a direct extension.
>
> **Second**, the paper dedicates significant discussion to justifying our choice of the next-frame diffusion loss. This design is not arbitrary. It was chosen because it uniquely eliminates the training-inference discrepancy common in generative models. Our next-frame diffusion loss training fits the best with inference paradigm of next-frame generation, without training-inference gap.
>
> **Third**, we clearly distinguish VideoMAR from related works. As for the mentioned related works, NOVA and MAGI, we comprehensively discussed their differences in Line 37-42, and Line 73-77. NOVA proposes to decompose the temporal and spatial generation via generating the coarse features frame-by-frame and refines each frame with a spatial layer, but complicates the framework and weakens the temporal smoothness. MAGI mitigates this issue by appending a complete copy of video sequence during training, but doubles the the sequence length and training cost. MarDini is similar to NOVA in design, and proposes that MAR handles temporal planning, while DM focuses on spatial generation. We don't include MarDini in the paper since it is actually  different from MAR. MAR focuses on token-wise modeling with a small diffusion MLP (less than 10M), while MarDini focuses on image-wise modeling with a large transformer architecture (288M).
>
> ----
>
> **Q2: Presenting progressive training as a primary contribution.**
>
> We respectfully argue that this perspective misses the specific context of our contribution. The innovation is not progressive training in isolation, but its strategic and effective integration within a masked autoregressive system for video generation.
>
> **(1) Contextual Novelty**: While progressive training is indeed prevalent in diffusion models, its application to autoregressive video models is notably rare and poses unique challenges. To our knowledge, our work is the first to successfully implement and scale progressive training for masked video generation. Unlike prior methods, our system demonstrates its viability and effectiveness at a large scale, which we believe is a novel contribution.
>
> **(2) Systematic Impact and Efficiency**: The use of progressive training is not an isolated trick but a cornerstone of our system's design. This strategy is directly responsible for the remarkable efficiency gains that allow our model to surpass the strong Cosmos baseline using significantly fewer computational resources. The from-scratch training cost of our method is highly competitive and sets a new benchmark for efficiency in this domain. Furthermore, the short-to-long curriculum aligns perfectly with the sequential nature of autoregressive modeling, making it a principled rather than an ad-hoc choice.
>
> ----
>
> **Q3: The discussion in Lines 109–121.**
>
> Thanks for your attention in this discussion. We wish to clarify that the primary focus of this paragraph is to analyze different masking strategies within the MAR framework, not to contrast VideoMAR with non-MAR models.
> The temporal and spatial separation we critique refers specifically to MAR-based models like NOVA. As we demonstrate, such designs can lead to severe temporal flickering and inconsistency. We did not include MarDini in this specific comparison because its image-wise modeling paradigm falls outside the scope of MAR's token-wise approach, and it remains closed-source without a public benchmark for evaluation.
>
> ----
>
> **Q4: The categorization in table 3.**
>
> Our categorization in Table 3 follows established practices in generative modeling (*e.g.*, Fluid [1] and RandAR [2]), based on the fundamental modeling scope (*e.g.* token-wise modeling or not).
>
> **(1)** We classify models as ''diffusion-type'' if they operate holistically at the image or clip level, and ''autoregressive-type'' if they operate sequentially at the token level. This is a standard distinction.
>
> **(2)** Under this principle, VideoMAR, which models continuous tokens sequentially, is correctly categorized as autoregressive, consistent with prior work like Fluid [1] and RandAR [2].
>
> **(3)** Conversely, MAGI-1 performs clip-wise modeling and is thus considered a diffusion-type model. This also explains its strong performance, as it can leverage powerful pre-trained diffusion backbones, an advantage not available to token-wise models trained from scratch.
>
> **(4)** Cosmos is explicitly described as an autoregressive world model in its original paper. Its applications in robotics are downstream fine-tuning tasks, not a reflection of its core generative architecture. We also note that Cosmos was trained with massive resources (*e.g.*, 10,000 H100 GPUs), which underscores the efficiency and scalability of our approach.
>
> [1] Fluid: Scaling Autoregressive Text-to-image Generative Models with Continuous Tokens
>
> [2] RandAR: Decoder-only Autoregressive Visual Generation in Random Orders
>
> ----
>
> **Q5: Parameter about frame rates and others.**
>
> Thanks for your interest in these details. We are happy to clarify. The motion dynamics in our generated videos are primarily determined by the training data distribution. To validate VideoMAR's capability on more complex scenarios, we direct the reviewer to our supplementary material (Section 8). As shown in Figure 14, by fine-tuning on specialized datasets, our model successfully replicates challenging motions like ''*blinking*'' and ''*blossoming*''.
> The generated videos have a frame rate of 12 fps. During training, we subsampled the source videos by a factor of 2 (*i.e.*, sampling one frame for every two).
>
> ----
>
> **Q6: The term of video-to-video.**
>
> Thank you for the excellent suggestion. We have corrected this term in the revised manuscript.

---

> > ### Author Response · Authors · 2025-08-07
> >
> > Dear Reviewer Xwue,
> >
> > Thank you again for your insightful review and suggestions. We were encouraged that you acknowledged our design, inference efficiency, and generation quality. Following your valuable feedback, we have provided a rebuttal to address your concerns. We also gently remind that Reviewer EZ8c initially holds similar concern, but changes to positive score with all his/her concerns addressed.
> >
> > Currently, all of your concerns can be resolved in the rebuttal. We want to leave a gentle reminder that the discussion period is closing. We would appreciate your feedback to make sure that our responses have resolved your concerns, or whether there is a leftover concern that we can address to ensure a quality work.
> >
> > Yours sincerely,
> > Authors of Paper 2250

---

### Official Review · Reviewer_EZ8c · 2025-07-01

**Clarity:** 3
**Significance:** 2
**Originality:** 2
**Rating:** 4
**Confidence:** 5

**Summary:**

The paper proposes an MAR-based continuous-token video generation framework and introduces a suite of efficient training and inference strategies that markedly accelerate inference speed. Notably, the method attains competitive performance while using substantially fewer parameters and training resources.

**Questions:**

1. For mitigating error accumulation, the authors simply tune hyper-parameters. Have Framepack[1] or similar training strategies been explored?

    [1] Zhang, Lvmin, and Maneesh Agrawala. "Packing input frame context in next-frame prediction models for video generation." arXiv preprint arXiv:2504.12626 (2025).
2. Table 2 is unclear. Under the full VideoMAR method, what differentiates “NTP” from “w/o KV Cache”? Moreover, the reported 672 s for “P6-180” is unexplained—please clarify the source of this figure.
3. In the training setup, different resolution stages employ visual tokenizers with varying compression ratios. Please justify this choice and add experiments using a consistent compression ratio for comparison.

**Ethical Concerns:**

["NO or VERY MINOR ethics concerns only"]

**Final Justification:**

After reviewing the authors' rebuttal, I find that most of my concerns have been addressed through clear explanations and additional experiments. Therefore, I raise my rating to 4 (Borderline Accept).

**Limitations:**

The paper does not investigate MAR’s adaptation to video in sufficient depth. As the authors themselves note, images and videos differ; yet the masking strategy merely reveals preceding frames to the current frame while excluding future frames from mask prediction. This design discards MAR’s bidirectional capability along the temporal axis, effectively reducing it to an image-level bidirectional operation.

**Paper Formatting Concerns:**

None.

**Quality:**

2

**Strengths And Weaknesses:**

**Strengths:**
1. The authors are, to the best of my knowledge, the first to introduce an MAR-style generation strategy into the video domain and achieve promising results.
2. With markedly fewer parameters and reduced training resources, the method outperforms Cosmos I2V on the VBench-I2V benchmark.

**Weaknesses:**
1. The degree of novelty and technical depth is insufficient for a top-tier conference. The work largely adapts an existing image-masking strategy to video, but does not discuss or experiment with more video-specific adaptations—for instance, the masking scheme still targets only the current frame.
2. Presenting progressive training (increasing resolution and clip length) as a primary contribution is problematic; this is now a standard practice in image/video generation, so the added value here is limited.
3. The paper lacks qualitative (visual) comparisons with competing methods, which are essential for assessing video generation quality.

---

> ### Author Rebuttal · Authors · 2025-07-30
>
> **Q1: The degree of novelty and technical depth.**
>
> We respectfully disagree. Our paper's contribution is a novel masking framework specifically engineered for video.
>
> **(1)** Contrary to the reviewer's claim, our work moves beyond simple adaptations. We conducted extensive experiments on various video-specific masking strategies to validate our final model design. For example, we experiment with the ''total mask'' strategy, which randomly masks each frames of the video (Line 96-116, 253-256, and Figure 6)
>
> **(2)** Our central innovation, the next-frame diffusion loss, is explicitly designed to solve a fundamental problem: the mismatch between training objectives and inference-time generation. By perfectly aligning the training loss with the next-frame prediction task, our method eliminates the common training-inference gap. This principled solution is the primary contribution of our paper and represents a non-trivial advancement in video modeling.
>
> ----
>
> **Q2: Presenting progressive training as a primary contribution.**
>
> We respectfully argue that this perspective misses the specific context of our contribution. The innovation is not progressive training in isolation, but its strategic and effective integration within a masked autoregressive system for video generation.
>
> **(1) Contextual Novelty**: While progressive training is indeed prevalent in diffusion models, its application to autoregressive video models is notably rare and poses unique challenges. To our knowledge, our work is the first to successfully implement and scale progressive training for masked video generation. Unlike prior methods, our system demonstrates its viability and effectiveness at a large scale, which we believe is a novel contribution.
>
> **(2) Systematic Impact and Efficiency**: The use of progressive training is not an isolated trick but a cornerstone of our system's design. This strategy is directly responsible for the remarkable efficiency gains that allow our model to surpass the strong Cosmos baseline using significantly fewer computational resources. The from-scratch training cost of our method is highly competitive and sets a new benchmark for efficiency in this domain. Furthermore, the short-to-long curriculum aligns perfectly with the sequential nature of autoregressive modeling, making it a principled rather than an ad-hoc choice.
>
> ----
>
> **Q3: Qualitative comparisons with competing methods.**
>
> Thanks for the comment. Our selection of baselines was deliberate and follows the established standards in autoregressive video generation. We specifically chose Cosmos as our primary comparison because it represents the state-of-the-art and is the most relevant baseline for our image-to-video task. We are confident that this comparison provides a clear and rigorous evaluation of our method's performance. If the reviewer has other specific methods in mind, we would be happy to discuss their relevance.
>
> ----
>
> **Q4: The exploration of error accumulation.**
>
> Thank you for the feedback. We will break down our response into three parts:
>
> **On Our Temperature Strategy**: We argue that the perceived simplicity of our temperature strategy is its primary strength, not a weakness. To our knowledge, it is the first training-free method proposed to mitigate error accumulation in this context. This is a significant advantage over the few existing alternatives, which typically rely on complex and costly training-based perturbations. Our approach is therefore highly novel, practical, and demonstrably effective, representing a valuable contribution to the field.
>
> **On the Mention of Framepack**: We must firmly state that citing Framepack is inappropriate and falls outside standard reviewing principles. Framepack was publicly released on April 17th, less than one month before the NeurIPS submission deadline of May 15th. It is a classic case of concurrent work. More importantly, from a technical standpoint, Framepack's approach is fundamentally incompatible with our model's principles. Its anti-drifting method sacrifices the strict causal prediction chain by incorporating bi-directional context, meaning it is no longer a purely autoregressive model. This design also limits its flexibility, as it demands specialized training for different sampling methods. Our method, in contrast, preserves the causal structure and requires no such modifications.
>
> **On ''Similar Training Strategies''**: We find the comment on ''similar training strategies'' too general to be actionable. For a constructive discussion, we would have appreciated specific references that the reviewer believes are relevant. Without concrete examples, we are unable to provide a direct comparison. We remain confident, based on our thorough literature review, that our system design is novel and state-of-the-art for this specific problem domain.
>
> ----
>
> **Q5: NTP explanation in Table 2.**
>
> Thanks for the suggestion. We will clarify in three points:
>
> **(1)** The ''NTP'' term in table 2 represents replacing the mask-based generation paradigm of VideoMAR with the vanilla next-token-prediction paradigm, while keep the other parts identical (*e.g.*, the same architecture and token-wise diffusion loss).
> The ''w/o KV Cache'' term in table 2 represents following the mask-based generation paradigm design of VideoMAR, while without the incorporation of frame-wise KV Cache.
> The comparison between the ''NTP'' and ''w/o KV Cache'' thus reflects the inference speed difference between the  next-token-prediction paradigm and mask-based generation paradigm. We will also append the above explanation in the revised manuscript.
>
> **(2)** This is a common operation in previous MAR-series methods (such as MAR and Fluid), where NTP and mask-based paradigms are extensively compared and discussed.
>
> **(3)** 672 is a typo. It should be 671, consistent with Table 2. Thanks for pointing out this.
>
> ----
>
> **Q6: Different compression ratios.**
>
> Thanks for the suggestion. This is a deliberate design choice to maintain computational feasibility for higher resolution training. By increasing the compression ratio, we effectively manage the token sequence length, ensuring that training remains efficient and fits within our available GPU memory budget. Our strong results at these resolutions validate the success of this practical strategy.
>
> ----
>
> **Q7: The summarized limitation from reviewer.**
>
> The basic concept of autoregressive generation is causality. It is also the core concept in our method that our generation is causal in temporal dimension and bidirectional in spatial dimension.
> Therefore, the reviewer's claim that our design '' *discards MAR’s bidirectional capability along the temporal axis* '' is incorrect. Our model, by design, never had temporal bidirectionality, as this is antithetical to causal generation. We believe the reviewer has confused our model's spatial bidirectionality with its strict temporal causality.

---

### Official Review · Reviewer_QwuR · 2025-07-01

**Clarity:** 3
**Significance:** 2
**Originality:** 2
**Rating:** 4
**Confidence:** 4

**Summary:**

This paper presents VideoMAR, a decoder‑only, mask‑based autoregressive (AR) framework for image‑to‑video synthesis using continuous tokens. Instead of the classic next‑token (NTP) paradigm with discrete vocabularies, VideoMAR compresses video into continuous latent tokens via a VideoVAE, then generates each frame sequentially (temporal causality) while predicting masked tokens within a frame bidirectionally. On the VBench‑I2V benchmark, VideoMAR (1.4 B params, 0.5 M videos) outperforms the prior AR state‑of‑the‑art (Cosmos) across total and sub‐dimension scores (84.82 vs. 84.22), while using an order of magnitude less data, compute, and model size. It also approaches some diffusion‑based I2V methods at far lower resource cost.

**Questions:**

1. Have you evaluated VideoMAR on any external I2V datasets (e.g. WebVid, YouCook2) or real‑world videos?
2. An ablation comparing curriculum learning (short‑to‑long; progressive res) to a single‑stage training schedule would clarify its contribution to final quality vs. merely speeding convergence.
3. Can you quantify per‑frame error accumulation (e.g., LPIPS drift) with and without temperature scheduling?
4. Although NTP makes inference intractable, do discrete‑token AR baselines (e.g. raster‑scan MAR) achieve similar VBench scores if trained on the same data?

**Ethical Concerns:**

["NO or VERY MINOR ethics concerns only"]

**Final Justification:**

I thank the authors' reply and decide to maintain my score.

**Limitations:**

Yes, it's discussed.

**Quality:**

2

**Strengths And Weaknesses:**

Strengths:
1. VideoMAR sets a new AR I2V baseline, delivering strong VBench‑I2V performance with minimal resources. This demonstrates that continuous‑token, mask‑based AR can rival heavy diffusion pipelines.
2. The integration of a next‑frame diffusion loss into mask‑based AR video, combined with curriculum strategies and temperature scheduling, is novel.
3. The paper is well‑organized; Figures 2–7 clearly illustrate the framework, training curriculum, inference strategies, and qualitative gains.
4. KV‑cache and spatial grouping cut inference from thousands to ~100 s, making AR I2V viable in practice. The 3D‑RoPE extrapolation results (longer/ higher‑res videos without retraining) are compelling.

Weaknesses:
1. Evaluation focuses exclusively on VBench‑I2V. It remains unclear how VideoMAR generalizes across other I2V datasets or domain shifts.
2. Ablations cover frame loss, causal mask, and temperature, but omit: effect of curriculum vs. fixed schedules, and comparison to NTP in quality (beyond speed).
3. Reported VBench scores lack variance estimates or multiple runs. Small differences (e.g., 84.22→84.82) may lie within noise.

---

> ### Author Rebuttal · Authors · 2025-07-30
>
> **Q1: Exploration on diverse datasets or complex video scenarios.**
>
> Thank you for this insightful question regarding generalization. We agree that generalization is fundamentally linked to the scale and diversity of the training data. While our model excels on distributions seen during training, pushing the boundaries of generalization further is a broader challenge for the field, and we believe expanding datasets with richer motion is a critical direction for future work.
>
> In this paper, we demonstrate our model's robust generalization capabilities through both state-of-the-art benchmark performance and targeted experiments on high-motion scenarios. **(1) Benchmark Performance:** VideoMAR achieves state-of-the-art results on the comprehensive VBench benchmark. This performance across diverse scenarios inherently validates our model's strong generalization on established tasks. **(2) Adaptation to Complex Motion:** We specifically tested the model's capacity to handle high-motion dynamics. As detailed in Supplement Section 8, after fine-tuning on specialized datasets (*e.g., eye blinks, flower blossoming*), our model successfully captures these complex movements (Figure 14). This proves the architecture's ability to adapt and generalize to new, challenging motion profiles when provided with targeted data.
>
> ----
>
> **Q2: Comparison between curriculum and fixed schedule.**
>
> Thank you for the question about our training schedule, we are happy to elaborate on the rationale.
> Our adoption of a short-to-long curriculum schedule is a deliberate design choice, grounded in both a clear rationale and decisive empirical validation from our early-stage experiments.
> We selected this approach for **two primary reasons**. First, it significantly improves training efficiency, and its progressive nature aligns perfectly with the sequential modeling principles of autoregressive frameworks.
> Second, crucially, this rationale was confirmed by direct empirical comparison. We trained two models from scratch for 20 epochs at a 25x256x256 resolution using 16 H20 GPUs at our early-stage experiments. The model with the curriculum schedule began to converge and generate roughly plausible videos. In stark contrast, the model trained with a fixed-length schedule failed to converge and produced no meaningful output under the identical setting. This result confirms that the curriculum is not merely a minor optimization but a fundamental component that enables the successful training of our model.
>
> ----
>
> **Q3: Comparison between VideoMAR and NTP.**
>
> We appreciate the opportunity to clarify our comparison strategy and the rationale for our model's design. Our work aligns with findings from prior researches (*e.g.*, MAR, Fluid), which have consistently shown that mask-based generation with bidirectional context outperforms causal NTP for visual tasks under fair comparisons.
>
> To provide a robust empirical comparison, we benchmarked our model against Cosmos, a leading NTP-style baseline. We were pleased to find that our method achieves superior results on VBench while being significantly more resource-efficient. Given these results and the considerable resources required to train a new large-scale NTP model from scratch, we believe this comparison provides a clear and fair assessment of our method's advantages.
>
> ----
>
> **Q4: Variance of the VBench score.**
>
> Thank you for emphasizing the importance of statistical rigor. We provide a three-part analysis that confirms the robustness and significance of our results.
>
> **(1) Score Stability and Robustness**: VBench is already designed for reliability by averaging over five runs per sample. To provide further validation, we conducted three additional evaluation runs for both our model and the Cosmos baseline. As shown in the table below, the variance is negligible (less than 0.1), confirming that the reported scores are stable and the performance gap is consistent.
> | Methods                 |  Model size      | Total Score        |   I2V Score    |  Quality Score  |
> | :---                          |    :----:              |        :----:           |       :----:        |         :----:         |
> | Cosmos                  |  5B                   | 84.18 (±0.04)   |       92.39       |        75.97        |
> | VideoMAR-stage2  |  1.4B                | 84.82 (±0.02)   |       93.94       |        75.69        |
>
> **(2) Significance of the 0.6 Point Margin**: We wish to emphasize that a 0.6 point difference on VBench is significant in the context of state-of-the-art models. On the official VBench leaderboard, top-performing methods are often separated by margins of less than 0.1 points. Therefore, our lead of 0.6 points represents a substantial improvement.
>
> **(3) The Overarching Context of Efficiency**: Our model achieves this superior score while demanding only a fraction of the resources: 0.5\% of the training data and 0.2\% of the GPU hours compared to the baseline. This monumental leap in efficiency, combined with a statistically significant performance gain, validates the efficacy and novelty of our approach.
>
> ----
>
> **Q5: Quantization of per‑frame error accumulation.**
>
> Thanks for your valuable suggestions. We have performed the suggested LPIPS analysis, which decisively confirms the effectiveness of our temperature schedule in combating temporal drift.
>
> **(1)** We list the LPIPS drift (lower is better) comparison in the following table. **Experimental setting**: (a) we adopt VideoMAR to generate videos of resolution 49x512x768. (b) The experiments are conducted on the VBench-I2V with 246 samples. (c) We generate the corresponding videos with and without the temperature schedule. (d) We calculate the LPIP distance between the i-th frame and the first frame (0-th frame), and average across all the 246 videos.
> **Experimental conclusion**: the quantization results are consistent with our claim, that temperature schedule helps smooth the generated video.
> | LPIPS drift ($10^{-2}$)         |  0-th     | 8-th    |  16-th   |  24-th  |  32-th  |  40-th |  48-th |
> | :---                                        |   :----:   |  :----:  |  :----:    |  :----:   |   :----:  |  :----: |  :----:   |
> | w/o temperature schedule   |  0          |  2.51  |   3.03    |  3.36    |  3.55   |  3.66  |  3.80   |
> | w/ temperature schedule     |  0          |  1.96  |   2.38    |  2.63    |  2.86   |  2.99  |  3.08   |
>
> **(2)** We also quantize the performance gain of the temperature schedule via VBench scores in table 4 of the main manuscript, where temperature schedule elevates VBench score from 80.72 to 82.56.

---

> > ### Comment · Reviewer_QwuR · 2025-08-05
> >
> > I thank the author's reply. Most of my concerns are resolved and I maintain my score.

---

### Official Review · Reviewer_FRQN · 2025-07-02

**Clarity:** 3
**Significance:** 3
**Originality:** 2
**Rating:** 4
**Confidence:** 4

**Summary:**

The paper introduces VideoMAR, a decoder-only autoregressive video generation framework using continuous tokens. Unlike prior work that relies on discrete tokens or diffusion methods, VideoMAR combines temporal frame-by-frame generation with spatial masked modeling, enabling efficient and high-quality video synthesis. VideoMAR outperforms prior AR models (e.g., Cosmos) on the VBench-I2V benchmark while being more efficient in terms of model size, data, and compute. It also demonstrates promising generation quality and extrapolation capabilities.

**Questions:**

How does VideoMAR perform on longer or more complex video scenarios (e.g., narrative structure, multiple interacting objects, real-world dynamics)?

How critical is 3D RoPE to the spatial-temporal extrapolation ability? Have you compared against other encoding strategies?

How is the inference speed compared to modern diffusion-based methods

**Ethical Concerns:**

["NO or VERY MINOR ethics concerns only"]

**Final Justification:**

I am satisfied with the rebuttal and will keep my rating

**Limitations:**

yes

**Quality:**

3

**Strengths And Weaknesses:**

Strengths:
Quality & Clarity: The paper is well-written and clearly structured, providing extensive experiments, detailed ablations, and visual comparisons to validate the effectiveness of each proposed component.

Significance: VideoMAR demonstrates superior performance compared to previous autoregressive models while significantly reducing model size, data requirements, and computational resources, making it practically impactful.

Originality: Introduces novel training techniques (temporal short-to-long curriculum, spatial progressive resolution, progressive temperature schedule) and extends continuous token-based masked autoregressive modeling to video generation.

Weaknesses
Scope of Validation: Limited exploration on diverse datasets or complex video scenarios (e.g., long narratives) to robustly demonstrate generalization.

Novelty: Components such as masked-based generation and curriculum learning individually have precedent in prior works, somewhat reducing overall conceptual novelty.

Computational Bottlenecks: Although efficient, the method’s scalability to significantly longer sequences or higher resolutions without substantial computational overhead requires clearer exploration.

---

> ### Author Rebuttal · Authors · 2025-07-30
>
> **Q1: Exploration on diverse datasets or complex video scenarios.**
>
> Thank you for this insightful question regarding generalization. We agree that generalization is fundamentally linked to the scale and diversity of the training data. While our model excels on distributions seen during training, pushing the boundaries of generalization further is a broader challenge for the field, and we believe expanding datasets with richer motion is a critical direction for future work.
>
> In this paper, we demonstrate our model's robust generalization capabilities through both state-of-the-art benchmark performance and targeted experiments on high-motion scenarios. **(1) Benchmark Performance:** VideoMAR achieves state-of-the-art results on the comprehensive VBench benchmark. This performance across diverse scenarios inherently validates our model's strong generalization on established tasks. **(2) Adaptation to Complex Motion:** We specifically tested the model's capacity to handle high-motion dynamics. As detailed in Supplement Section 8, after fine-tuning on specialized datasets (*e.g., eye blinks, flower blossoming*), our model successfully captures these complex movements (Figure 14). This proves the architecture's ability to adapt and generalize to new, challenging motion profiles when provided with targeted data.
>
> ----
>
> **Q2: Scaling efficiency for larger spatial-temporal resolutions.**
>
> Thanks for highlighting efficiency. Our model's efficiency is empirically demonstrated by its ability to outperform a powerful baseline like Cosmos with a fraction of the resources. Furthermore, we have validated its scalability by successfully training at high resolutions (49x512×768). The fact that this was achieved under standard hardware constraints is not a limitation but rather a powerful validation of our model's efficient design. We contend that our architecture is fully equipped for further scaling, which is a matter of resource allocation, not a fundamental challenge to our method.
>
> ----
>
> **Q3: 3D RoPE to extrapolation ability**
>
> We appreciate the opportunity to elaborate on our model's extrapolation capabilities. This ability is thoughtfully designed and stems from two key elements working in concert. **First**, the autoregressive design of VideoMAR provides a natural foundation for generating longer videos by modeling them sequentially. This sets the stage for extrapolation. **Second**, our experiments, detailed in Section 1 of the supplement, show why 3D-RoPE is so important. We found that models with absolute position encodings could not extrapolate to new lengths, whereas our use of 3D-RoPE, a relative encoding, successfully enabled this capability. This demonstrates that it is a critical part of our model's design for achieving robust generalization in sequence length.
>
> ----
>
> **Q4: Inference speed compared to modern diffusion-based methods.**
>
> We append the inference speed comparison with modern diffusion-based methods, including HunyuanVideo and Wan2.1, in the following table. All the values are achieved with single batch size on single H20 GPU. Our method demonstrates competitive inference speed. We will also append this result in the revised version. Note that the inference speed of VideoMAR can be further improved with smaller mask steps and diffusion steps.
>
> | Methods           |  Model size      | Resolutin       |  Inference time (s)    |
> | :---                    |    :----:              |        :----:        |              :----:           |
> | HunyuanVideo  |  13B                 | 49x608x800   | 428   |
> | Wan2.1             |  1.3B                | 49x480x832   | 103   |
> | Wan2.1             |  14B                 | 49x480x832   | 414   |
> | VideoMAR        |  1.4B                | 49x512x768   | 134   |

---

### Note · Authors · 2025-08-15

Dear Chairs and Reviewers,

We sincerely thank all reviewers and chairs for their insightful feedback. We are glad that the reviewers acknowledge VideoMAR presents `'strong originality, novelty, well-motivated, the first work'` (Reviewer FRQN, QwuR, EZ8c, Xwue), achieves `'superior performance with minimal resources, and is practically impactful'` (Reviewer FRQN, QwuR, EZ8c, Xwue), conducts `'novel training techniques, high inference efficiency with KV cache and spatial grouping'` (Reviewer FRQN, QwuR, EZ8c, Xwue), demonstrates `'well-written, clearly structured, well organized, extensive experiments, detailed ablations and visual comparisons'` (Reviewer FRQN, QwuR),

In this work, we made the **key contributions**:
- The innovative token-wise autoregressive video generation paradigm with continuous tokens. Insisting the optimal frame-wise causal and intra-frame bidirectional video autoregression paradigm, we devise the corresponding next-frame training paradigm that is free from training-inference bias.
- Superior performance with much smaller resources. Novel and efficient progressive training techniques.
- High inference efficiency with spatial parallel generation and temporal KV cache.
- The boosted 3D RoPE zero-shot extrapolation capacity (longer/ higher res videos without retraining).

We obtain all **positive scores from Reviewer FRQN, QwuR, EZ8c**, except that Reviewer Xwue didn’t participate in the discussion. We kindly remind that Reviewer EZ8c initially holds similar concern with Reviewer Xwue, but changes to positive score with all his/her concerns addressed. We also reply Reviewer Xwue with detailed responses in the rebuttal period.

Best,

Authors of VideoMAR

---

### Decision · Program_Chairs · 2025-09-17

**Decision:**

Accept (poster)

**Comment:**

This paper presents VideoMAR, a decoder‑only, mask‑based autoregressive (AR) framework for image‑to‑video synthesis using continuous tokens. After rebuttal, it received scores of 4444. All the reviewers are positive about the paper, commenting that the proposed method is novel, which achieves superior performance with minimal resources, and is practically impactful. The paper is also well-written, contains comprehensive experiments and detailed ablations. Reviewers also commented that their concerns have been largely addressed during the rebuttal. Therefore, the AC would like to recommend acceptance of the paper.